# Rigid firing sequences undermine spatial memory codes in a neurodegenerative mouse model

**Jingheng Cheng[1], Daoyun Ji[1,2]\***

[1]Department of Molecular and Cellular Biology, Baylor College of Medicine, Houston, United States; [2]Department of Neuroscience, Baylor College of Medicine, Houston, United States

**Abstract** Hippocampal neurons encode spatial memories by firing at specific locations. As the animal traverses a spatial trajectory, individual locations along the trajectory activate these neurons in a unique firing sequence, which yields a memory code representing the trajectory. How this type of memory code is altered in dementia-producing neurodegenerative disorders is unknown. Here we show that in transgenic rTg4510 mice, a model of tauopathies including Alzheimer's disease, hippocampal neurons did not fire at specific locations, yet displayed robust firing sequences as animals run along familiar or novel trajectories. The sequences seen on the trajectories also appeared during free exploration of open spaces. The spatially dissociated firing sequences suggest that hippocampal neurons in the transgenic mice are not primarily driven by external space but by internally generated brain activities. We propose that tau pathology and/or neurodegeneration renders hippocampal circuits overwhelmed by internal information and therefore prevents them from encoding spatial memories.

**\*For correspondence:** dji@bcm.edu

**Competing interests:** The authors declare that no competing interests exist.

## Introduction

Alzheimer's disease (AD) is behaviorally characterized by cognitive declines such as memory loss (*Carlesimo and Oscar-Berman, 1992*; *Salmon and Bondi, 2009*), but pathologically defined by β-amyloid plaques and tau pathology including neurofibrillary tangles and tau-mediated neurodegeneration (*Wenk, 2003*; *Ashe and Zahs, 2010*). A current challenge is how to bridge between these two very different aspects of AD. Because memory deficits are necessarily produced by the memory-processing neural circuits in vivo (*Dickerson and Eichenbaum, 2010*), a key step toward this challenge is to understand the functional alterations in the memory circuits, including the hippocampus (HP), of the living brain with ongoing pathogenesis. Both the β-amyloid and tau pathologies are likely involved in the memory loss (*Chapman et al., 1999*; *Yu et al., 2001*; *Cacucci et al., 2008*; *Ashe and Zahs, 2010*; *Palop and Mucke, 2010*). In particular, tau pathology and subsequent neurodegeneration play a key role in mediating the memory symptoms seen at a mature stage of AD (*Ashe and Zahs, 2010*). Currently, it is unknown how the mnemonic functions of HP are altered by tau pathology in vivo.

One well-studied function of HP in vivo is the coding of spatial memories. It is believed that spatial memories in both rodents and humans are encoded by HP place cells that fire at one or a few specific locations of a given space (place fields) (*O'Keefe and Dostrovsky, 1971*; *Wilson and McNaughton, 1993*; *Burgess and O'Keefe, 2003*; *Ekstrom et al., 2003*). When the animal runs along a novel spatial trajectory, external sensory input at individual locations along the trajectory drives place cells to fire one after another in a unique sequence, which yields a memory code for the trajectory (*Wilson and McNaughton, 1993*; *Harris et al., 2003*; *Dragoi and Buzsaki, 2006*). After the novel experience, the sequence becomes internally established and can be reactivated during later memory recall and/or

**eLife digest** Patients with Alzheimer's disease often forget where they have been or who they have just met. This happens because the neurons in those areas of the brain where memories are processed are dying. Indeed, by the time Alzheimer's disease has been diagnosed, many of the neurons in these regions have already died. The symptoms of Alzheimer's disease are then produced by the remaining neurons. However, the reasons why the remaining neurons cannot make new memories are unknown.

In normal mice the neurons in the hippocampus, a part of the brain that is important for memory, are called 'place cells' because they are turned on when the mouse is in a specific place. As a consequence, when the mice moves around, different neurons are turned on one by one, and this sequence of activation is believed to be a memory code that represents the places the animal has travelled.

Cheng and Ji have explored this phenomenon in mice that have been genetically engineered so that their neurons contain structures called 'tau tangles' that are thought to be involved in the death of neurons. More importantly, these transgenic mice suffer age-dependent neuron loss in a way that is similarly to people with Alzheimer's disease.

Cheng and Ji implanted tiny sensors into the hippocampus of these mice, and used these sensors to monitor the activity of the remaining hippocampal neurons as the mice moved around while searching for food. They found that the neurons were activated almost everywhere, which indicates that the hippocampal neurons in transgenic mice are no longer working as place cells. However, these neurons were still activated one by one in robust sequences. Moreover, the sequences generated by the transgenic mice were the same in many different surroundings, which suggests that these sequences are not memory codes of the animal's current surroundings. Cheng and Ji propose that the sequences reflect existing memories already stored in the brain, which would suggest that Alzheimer's patients cannot form new memories because the brain is preoccupied by old memories, and thus fails to store the new information that is coming in from the outside world.

consolidation with or without partial external input (*Buzsaki, 1989*; *Wilson and McNaughton, 1994*; *Skaggs and McNaughton, 1996*; *Nadasdy et al., 1999*; *Lee and Wilson, 2002*; *Foster and Wilson, 2006*; *Diba and Buzsaki, 2007*; *Ji and Wilson, 2007*; *Gelbard-Sagiv et al., 2008*; *Davidson et al., 2009*; *Karlsson and Frank, 2009*; *Carr et al., 2011*). Therefore, in theory, on any given trajectory HP could either retrieve stored internal sequences, form new ones, or combine both, based on the external input that HP receives from the environment. It has been proposed that the interplay between internal and external inputs is important to memory processing in HP and abnormal interaction may lead to memory interference and/or intrusion (*Nakazawa et al., 2002*; *McHugh et al., 2007*; *Colgin et al., 2008*).

The availability of tauopathy animal models provides an opportunity to study how tau pathology alters HP mnemonic functions in vivo. To this end, we set out to examine the spatial memory code in the transgenic rTg4510 (Tau) mouse. This model was chosen because its well-characterized pathological and behavioral phenotypes mimic many key features of tau pathology in AD (*Ramsden et al., 2005*; *Santacruz et al., 2005*; *Berger et al., 2007*). In Tau mice, the overexpression of a mutated version (P301L) of the human tau gene restricted in the forebrain leads to age-dependent spatial memory deficits, tau neurofibrillary tangles, and neuronal loss. In this study, we recorded neurons in the CA1 region of HP using the tetrode recording technique (*Wilson and McNaughton, 1993*; *Buzsaki, 2004*; *Ji and Wilson, 2007*) from 7 to 9 month old Tau mice and their control littermates while they performed spatial navigation tasks. We found that HP neurons in Tau mice fail to fire at specific spatial locations and yet maintain robust firing sequences, suggesting that non-spatial, internal brain activities dominate the HP circuits and prevent them encoding external space.

## Results

We recorded a total of 497 CA1 neurons from eight Tau and six wildtype (WT) control mice, all at 7–9 months old (*Table 1*). Confirming previous studies (*Ramsden et al., 2005*; *Santacruz et al., 2005*), there was massive neurodegeneration in Tau mice at this age and the thickness of the CA1 pyramidal layer in HP of Tau mice was on average only 48% of that of WT mice (*Figure 1* and *Table 1*).

**Table 1.** Details of the recorded mice

| Animal name | Genotype | Sex | Age (m) | Brain weight (g) | CA1 thickness (μm) | Cortex thickness (μm) | $N_f$ | $N_n$ |
|---|---|---|---|---|---|---|---|---|
| AN1 | WT | M | 8–9 | NA | 56 | 800 | 7 | 3 |
| AN4 | WT | M | 8–9 | 0.350 | 51 | 790 | 16 | 0 |
| AN18 | WT | M | 7–8 | 0.334 | 55 | 760 | 8 | 10 |
| AN19 | WT | F | 7–8 | 0.324 | 51 | 690 | 41 | 30 |
| AN20 | WT | F | 7–8 | 0.392 | 51 | 690 | 16 | 37 |
| AN21 | WT | F | 8–9 | 0.328 | 51 | 690 | 31 | 34 |
| AN10 | +/− | M | 8–9 | 0.364 | NA | NA | 42 | 33 |
| AN11 | +/− | M | 7–8 | 0.356 | 53 | 760 | 10 | 15 |
| AN12 | −/+ | F | 7–8 | 0.310 | 50 | 800 | 12 | 18 |
| AN16 | −/+ | M | 8–9 | 0.332 | 60 | 800 | 15 | 18 |
| AN5 | Tau | F | 8–9 | 0.192 | 26 | 580 | 17 | 15 |
| AN13 | Tau | F | 8–9 | 0.226 | 22 | 580 | 14 | 10 |
| AN14 | Tau | F | 8–9 | 0.236 | 27 | 580 | 24 | 25 |
| AN15 | Tau | M | 8–9 | 0.214 | 26 | 580 | 10 | 18 |
| AN22 | Tau | M | 7–8 | NA | 28 | 550 | 20 | 10 |
| AN23 | Tau | M | 7–8 | 0.198 | 24 | 550 | 7 | 10 |
| AN24 | Tau | F | 7–8 | 0.232 | 27 | 550 | 14 | 14 |
| AN25 | Tau | M | 7–8 | 0.226 | 26 | NA | 29 | 27 |
| | | | | | | Total | 333 | 327 |

Genotype: see 'Materials and methods'. Age: age of the mouse in months when data were recorded. Brain weight: weight of the brain with the cerebellum and olfactory bulb removed. CA1 thickness: thickness of the pyramidal cell layer in the dorsal CA1 of HP; cortex thickness: thickness of the parietal cortex above the dorsal CA1; measurements were made in the coronal sections that contained the most dorsal point of CA1, about the section at Bregma -2.06 (Figure 48) of the Franklin and Paxinos mouse atlas (***Franklin and Paxinos, 2008***). $N_f$/$N_n$: number of CA1 neurons recorded in the familiar/novel room. NA: number not available. Note the lower brain weight, the thinner CA1 pyramidal layer, and the thinner cortex in Tau mice.

## CA1 neurons in Tau mice did not fire at specific locations on a familiar track

First, we asked whether CA1 place cell activities were altered in Tau mice when they ran along a familiar rectangular track (***Figure 2A***) two sessions a day, 15 min each session. In each session, the animals ran back and forth (two trajectories) and repeated 10–40 laps on each trajectory. We analyzed 87 putative pyramidal neurons from WT mice (WT neurons) and 93 from Tau mice (Tau neurons) that were active on at least one trajectory in at least one session. The median firing rate of Tau neurons during the track running sessions was not significantly different from that of WT neurons (median and [10% 90%] range rate: WT 1.21 [0.43 3.60] Hz, Tau 0.90 [0.41 2.75] Hz, p=0.08, *ranksum*-test, same below unless specified otherwise).

As expected, WT neurons fired spikes at stable, specific locations in every lap that the animal traversed a trajectory. In contrast, spikes of Tau neurons were not confined to specific locations (***Figure 2B***). Close visual inspections revealed that Tau neurons' spikes were still specific to one or a few locations in each individual lap, but the locations were changed from lap to lap (***Figure 2B***). We described a neuron's firing profile on a trajectory by a rate curve (the two trajectories were analyzed separately for each neuron), which was the neuron's firing rate as a function of the position along the trajectory. The rate curve was computed for every lap and then averaged across all laps. Whereas the averaged rate curves of typical WT neurons clearly showed one or a few peaks indicative of well-defined place fields, those of typical Tau neurons did not show such prominent peaks (***Figure 2B***, 'bottom curves'). We quantified the location-specificity using spatial information (SI) (***Skaggs et al., 1993***). SI was computed for each neuron active on each trajectory at two levels, a lap SI to measure the location-specificity at

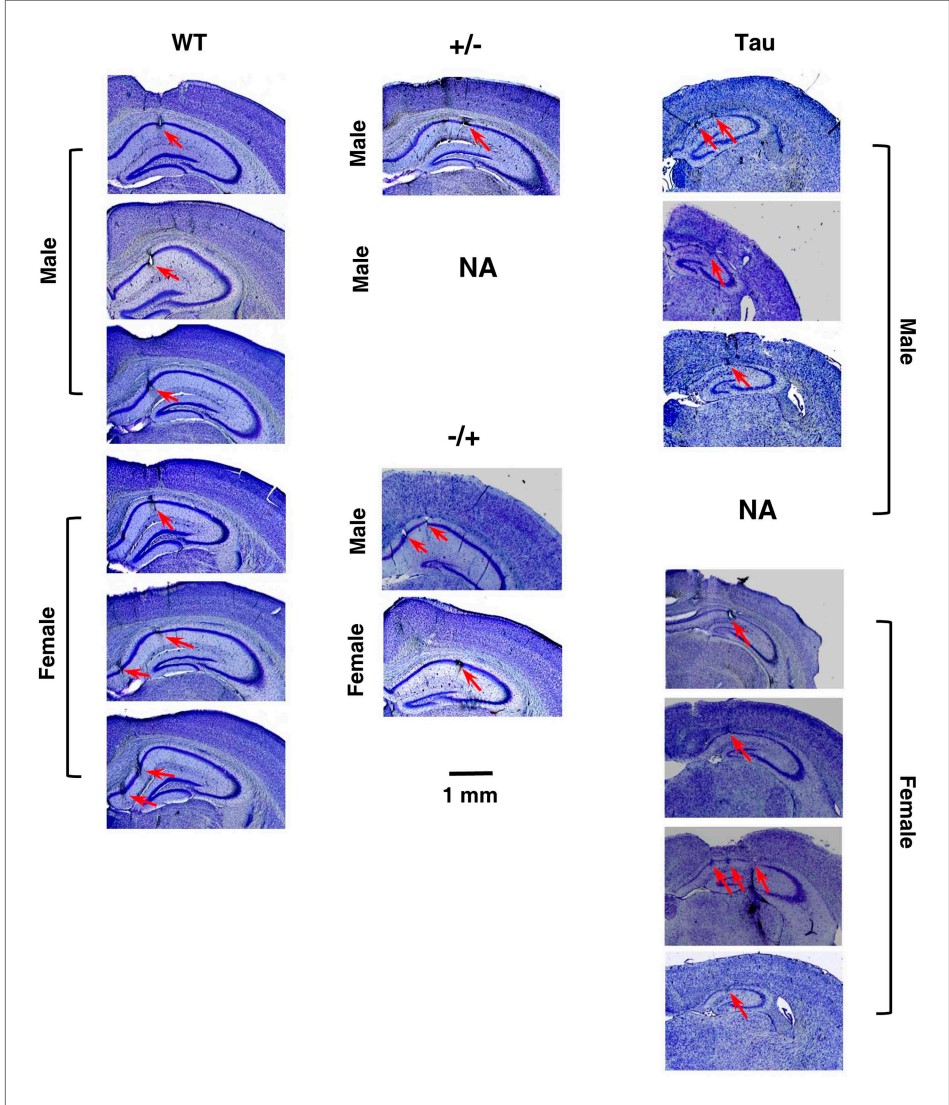

**Figure 1**. Histological confirmation of recording sites and neurodegeneration in Tau mice. The images are cresyl violet stained coronal brain sections, showing (1) the recording sites at the CA1 pyramidal cell layer of HP and (2) the prominent cell loss in Tau mice, compared with WT mice and mice with other two control genotypes (+/− and −/+). One section per recorded mouse is shown. One +/− mouse was not available for sectioning (NA). The recorded side of the brain in one Tau mouse was damaged during sectioning (NA). Recording sites at CA1 in these two mice were identified by electrode depths and the presence of sharp-wave ripple complexes (*Csicsvari et al., 2000*). Scale bar applies to all sections. *Arrows*: recording sites. Note the smaller hippocampi and thinner CA1 pyramidal cell layers in Tau mice.

individual lap level and a trajectory SI to measure the overall location-specificity across all laps based on its averaged rate curve. The median lap SI of WT neurons was only slightly (21%) greater than that of Tau neurons (*Figure 2C*; WT 2.03 [1.00 3.09] bits/spike, $N = 263$; Tau 1.68 [0.84 2.34] bits/spike, $N = 262$; $p=7.7 \times 10^{-11}$), but the median trajectory SI of WT neurons was much greater (180%) than that of Tau neurons (*Figure 2D*; WT: 1.29 [0.55 2.61] bits/spike; Tau: 0.46 [0.15 1.29] bits/spike; $p=5.3 \times 10^{-39}$). We also quantified the lap-to-lap location change by rate-stability, defined as the average correlation coefficient between any two laps' rate curves. Tau neurons showed much lower rate-stability than WT neurons (*Figure 2E*; WT: 0.66 [0.30 0.86]; Tau: 0.075 [0.0042 0.34]; $p=1.7 \times 10^{-70}$). These results reveal that Tau neurons lost their overall location-specificity, mainly due to unstable firing locations.

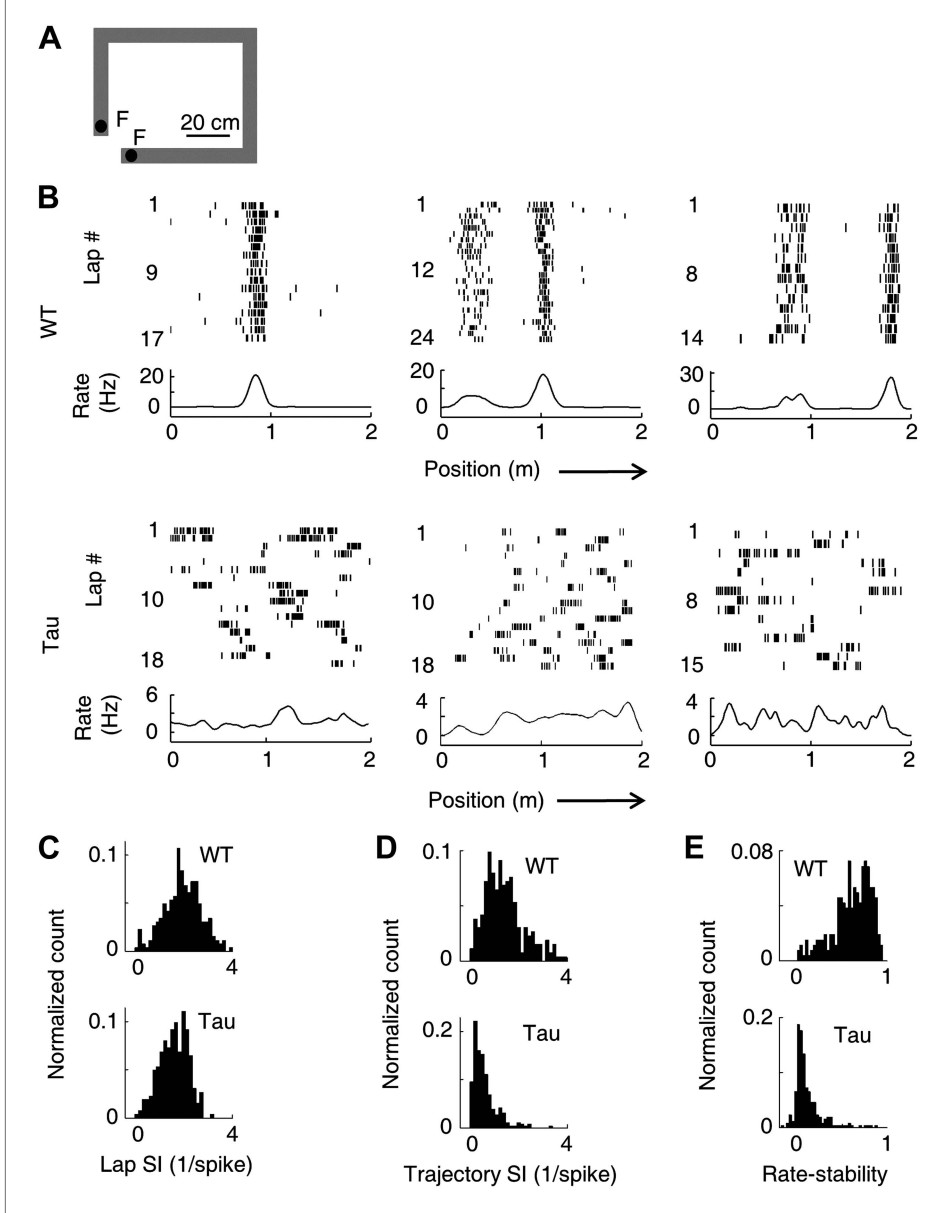

**Figure 2**. Unstable firing locations of Tau neurons led to the loss of overall location-specificity on familiar trajectories. (**A**) The rectangular familiar track. *F*: food wells. (**B**) Lap-by-lap spike raster of three WT and three Tau neurons, each from a different animal, on one trajectory of the track. The trajectories were linearized and plotted as the x-axis (*arrows*: running directions). Each tick represents a spike. *Bottom curves*: firing rates on the trajectories averaged across all laps. Note the unstable firing locations of the Tau neurons. (**C**)–(**E**) Distribution of lap SI (**C**), trajectory SI (**D**), and rate-stability (**E**) of WT and Tau neurons (see text for definitions). Plots are histograms normalized by total numbers of samples, each computed for one neuron on one trajectory.

## CA1 neurons in Tau mice maintained robust firing sequences on the familiar track

Because WT neurons fired at stable locations, they fired one after another with a stable, position-locked sequence in every lap of a trajectory (*Figure 3A*). To our surprise, despite their unstable firing locations, Tau neurons still maintained stable firing sequences, that is, they fired with consistent orders across laps (*Figure 3A*).

To quantitatively demonstrate this, we analyzed the high-order (≥4 neurons) firing sequences (*Lee and Wilson, 2002*; *Ji and Wilson, 2007*) of WT/Tau neurons on the familiar track. We first assigned

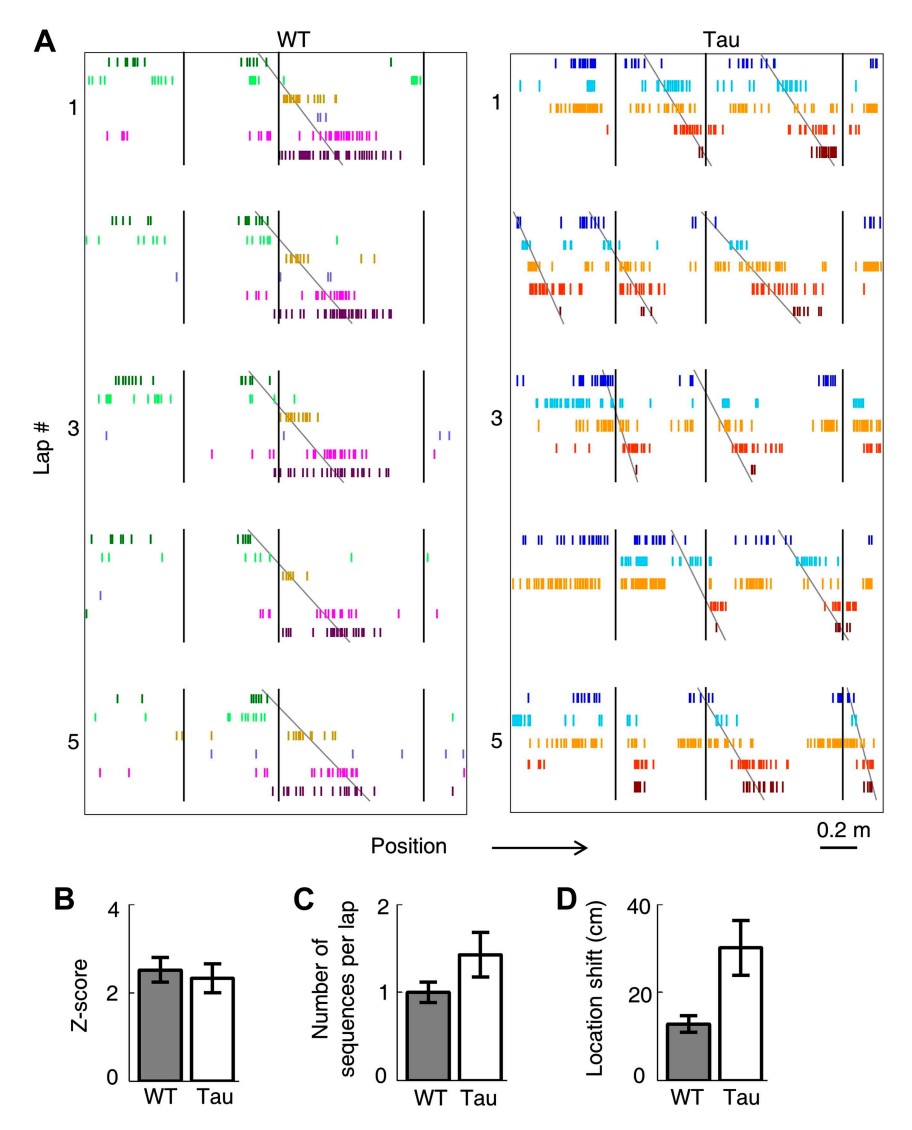

**Figure 3**. Tau neurons maintained stable firing sequences on familiar trajectories. (**A**) Lap-by-lap spike raster of six WT neurons and that of five Tau neurons on a trajectory (*arrow*: running direction). Each color-coded row within a lap represents a neuron. Each tick represents a spike. Only five laps are shown. *Angled lines*: detected firing sequences that matched with the templates derived on the trajectories (*see Figure 3—figure supplements 1 and 2* for details of template generation and sequence detection). *Vertical lines*: landmark positions (three corners of the track in **Figure 2A**) along the trajectory. Note that the Tau neurons shifted their firing locations, but maintained the same firing sequences across the laps. See more data from more animals with more laps in **Figure 3—figure supplement 3**. (**B**) Mean number of detected sequences (in Z-score) on the track trajectories in WT and Tau mice. The robustness of this sequence detection result is examined in **Figure 3—figure supplement 4**. (**C** and **D**) Mean number of sequences per lap (**C**) and mean location shift (**D**) for the detected sequences in WT and Tau mice.

The following figure supplements are available for figure 3:

**Figure supplement 1**. Deriving template sequences based on pair-wise cross-correlations.

**Figure supplement 2**. Detecting matches with a template sequence.

**Figure supplement 3**. More examples of WT and Tau firing sequences on familiar trajectories.

**Figure supplement 4**. The robustness of the sequence detection result.

letter identities to those neurons active on a trajectory. We arranged pairs of neurons according to their firing orders identified by their cross-correlations. We then derived a template sequence for the trajectory that agreed with all the ordered pairs. For example, for the five Tau neurons shown in *Figure 3A*, the cross-correlation between the second (assigned as neuron *A*) and the fourth (neuron *H*) had a positive peak at 0.8 s. We ordered them as *AH*. By ordering all pairs among the five neurons in this manner, we derived a template sequence *FABHD* for this trajectory. A total of 13 template sequences were derived from WT mice and 8 from Tau mice (*Table 2*). For a template sequence on a trajectory, we then detected the firing sequences in the spiking patterns of all laps that matched with the template. To evaluate the significance of the detection, we generated 1000 shuffled copies of the spike patterns by randomly swapping neuron identities and then detected the number of sequences in each copy. The number of detected sequences in the real spike patterns was expressed as a *Z*-score computed from the mean and standard deviation of the shuffling-generated counts. We found a significant number of detected sequences for 9 out of the 13 WT templates and 6 of the 8 Tau templates (p<0.05, *Z*-test; *Table 2*). The mean (±SEM) *Z*-scores for all WT and all Tau templates were similarly high (*Figure 3B*; WT: 2.5 ± 0.3; Tau: 2.3 ± 0.3; p=0.67, one tailed *t*-test, same below unless specified otherwise). Both WT and Tau template sequences were detected in almost every lap (*Figure 3A*). Although it appeared that Tau sequences occurred more frequently than WT sequences in a given lap (*Figure 3A*), this difference was not significant (*Figure 3C*; WT: 1.0 ± 0.1 times per lap; Tau: 1.4 ± 0.3 times per lap; p=0.10). These results show that Tau neurons, like WT neurons, formed robust firing sequences.

**Table 2.** High-order sequence analysis results for individual template sequences derived on the familiar trajectories

| Template sequence | Animal name | Genotype | Self traj. | Z (S1T1) | Z (S1T2) | Z (S2T1) | Z (S2T2) | Z (O1) | Z (O2) |
|---|---|---|---|---|---|---|---|---|---|
| AGFHJ | AN4 | WT | S2T2 | 0.01 | 2.8 | −0.4 | **3.2** | NA | NA |
| ZSBPNA | AN19 | WT | S1T1 | **2.6** | −0.7 | 1.4 | 0.3 | 0.8 | −1.7 |
| JHLKZE | AN19 | WT | S1T2 | 1.6 | **2.7** | −0.4 | 1.8 | −0.1 | 0.6 |
| IYOBZENA | AN19 | WT | S2T1 | 1.9 | 2.3 | **3.5** | 1.3 | 0.7 | 2.7 |
| ZSaFbB | AN19 | WT | S2T2 | −0.3 | 1.7 | 0.9 | **1.9** | 1.9 | 0.3 |
| GHKJBDF | AN20 | WT | S1T1 | **1.6** | 1.2 | 2.4 | 1.0 | −1.8 | 0.9 |
| JHAKC | AN20 | WT | S1T2 | −0.6 | **1.3** | −0.3 | 2.1 | 0.02 | −0.7 |
| GHCKJBDF | AN20 | WT | S2T1 | 1.8 | 0.3 | **1.4** | 1.3 | −0.6 | 1.4 |
| GJAHLC | AN20 | WT | S2T2 | 0.5 | 1.1 | 0.4 | **0.8** | −1.8 | −0.3 |
| XIGQOZ | AN21 | WT | S1T1 | **4.0** | −1.2 | 3.6 | −1.0 | −2.0 | −1.0 |
| IWQVHP | AN21 | WT | S1T2 | −0.8 | **3.0** | −1.2 | 3.3 | 0.2 | 0.0 |
| GOZTHR | AN21 | WT | S2T1 | 1.9 | 0.5 | **3.6** | 0.3 | 1.0 | −1.5 |
| OFINKDV | AN21 | WT | S2T2 | −0.9 | 3.8 | 0.5 | **3.3** | −0.5 | −1.4 |
| ACHFDG | AN5 | Tau | S2T1 | 3.0 | −0.4 | **2.5** | −0.06 | −1.2 | −1.6 |
| FABHD | AN13 | Tau | S1T1 | **4.0** | −0.1 | 2.9 | 2.0 | 2.1 | 2.9 |
| BFAHD | AN13 | Tau | S2T1 | 1.7 | 1.1 | **2.1** | −0.9 | 0.04 | 0.5 |
| FEBA | AN15 | Tau | S2T1 | 1.0 | 0.9 | **3.2** | −0.4 | 2.6 | 1.9 |
| ILFBK | AN22 | Tau | S1T1 | **1.4** | 0.5 | −0.8 | −0.8 | 1.2 | −0.5 |
| FLBDIC | AN22 | Tau | S2T1 | 0.1 | 1.4 | **2.2** | 0.7 | 2.2 | 2.9 |
| PKLEDO | AN25 | Tau | S1T2 | 1.1 | **1.9** | 0.4 | 3.3 | 2.4 | 2.4 |
| PKTLEOMA | AN25 | Tau | S2T2 | 0.7 | 1.8 | 0.0 | **1.3** | 3.4 | 1.9 |

Template sequence: each letter (case-sensitive) represents a cell; same letters across different templates but within the same animal represent the same cells. Self traj.: The trajectory where the template was derived. *Z*: Number of detected matches (in *Z*-score) with a template sequence on a trajectory or in an open box session. S1T1: Trajectory one of session one, S1T2: Trajectory two of session one, so forth; O1: First open box session, so forth; numbers on self-trajectories are in bold; numbers that are statistically significant (p<0.05) are in red. NA: not available (not all experiments included two trajectory running sessions and two open box sessions).

However, there was a major difference between WT and Tau sequences. Whereas WT template sequences predominantly occurred at similar locations across laps, Tau sequences shifted their locations from lap to lap (*Figure 3A*). The mean location shift of the detected Tau template sequences was significantly greater than that of WT sequences (*Figure 3D*; WT: 12.7 ± 1.9 cm; Tau: 31.3 ± 6.2 cm; p=0.0035). Therefore, unlike WT sequences, Tau sequences were not anchored to specific spatial locations.

### The firing sequences of CA1 neurons in Tau mice seen on the familiar track also appeared during free exploration of a familiar open box

We have shown that CA1 neurons maintained robust firing sequences during track running, even though they did not fire at specific locations. We next asked whether this was true during a very different behavior in a different familiar space. After completing the track running sessions, we kept recording the same CA1 neurons while the mice freely explored a familiar open box (*Figure 4A*) for one to two sessions, 15 min each session.

We analyzed 74 WT and 68 Tau neurons that were active in at least one open box session. The median firing rate of these Tau neurons was lower than that of WT neurons (WT 1.3 [0.5 3.8] Hz, Tau 0.9 [0.5 2.7] Hz, p=0.0001), which would predict a greater location-specificity of Tau neurons. However, we observed the opposite. As illustrated by their rate maps (*Figure 4B*), firing rate color-plotted at each position of the two dimensional (2D) open box, firing activities of Tau neurons covered a much larger portion of the box (meaning less location-specific) than those of WT neurons. The location-specificity in the box was quantified by SI in the 2D open space (open SI) for each neuron active in each open box session. The median open SI of WT neurons was significantly greater than that of Tau neurons (*Figure 4C*; WT: 0.24 [0.04 0.76], $N$ = 131; Tau: 0.11 [0.04 0.34], $N$ = 119; p=1.9 × 10$^{-7}$). The result indicates that Tau neurons fired with low location-specificity in the familiar open box.

We then examined whether there were organized firing sequences of multiple neurons during the open box sessions. Because of the random nature of the free exploration behavior, unlike the repeated

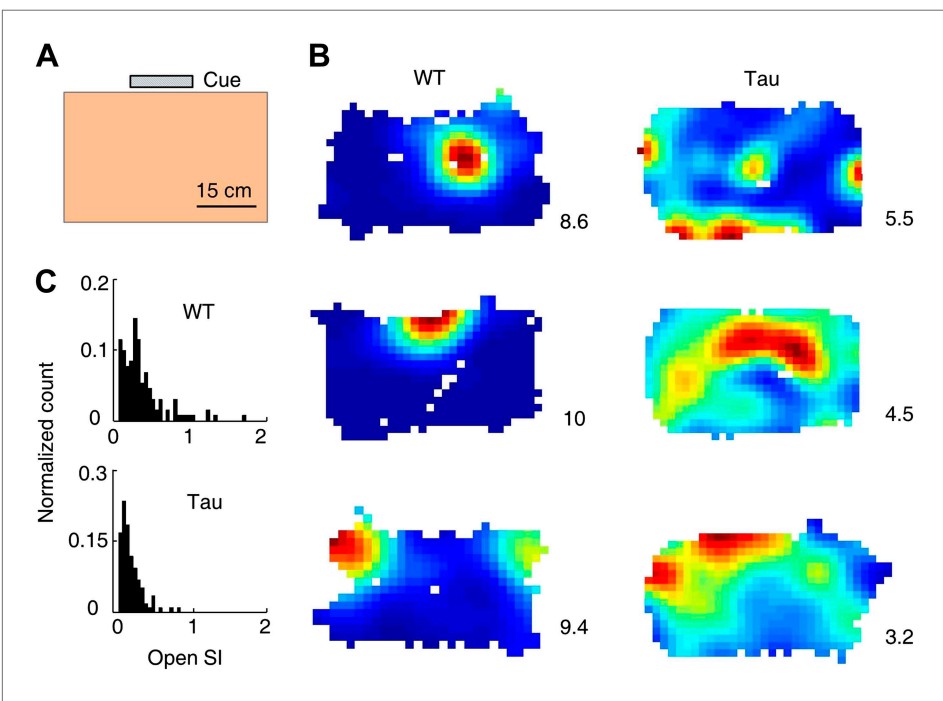

**Figure 4**. Tau neurons fired with low location-specificity in a familiar open box. (**A**) The open box with its interior color and cue card (*Cue*) shown. (**B**) Color-coded firing rate maps of three WT and three Tau neurons, each from a different animal, in the open box. *Numbers*: peak (red/black) rates in Hertz. Note the broader firing areas of Tau neurons than those of WT neurons. (**C**) Distribution of open SI of WT and Tau neurons. Plots are histograms normalized by total numbers of samples, each computed for one neuron in one open box session.

trajectory running on the track, we did not expect repetitive high-order firing sequences to appear in the open box. However, when we arranged the neurons according to their firing orders in the template sequences derived during track running, it became apparent that Tau neuron sequences, but not WT sequences, during the track running also appeared frequently during the open box sessions (*Figure 5A*). We quantified this observation by searching through the spike patterns of open box sessions for sequences that matched with the trajectory template sequences. Here we refer 'a search' as to matching one open box session to one trajectory template. We found a significant number of sequences in 10 out of the 16 searches in Tau mice, whereas only in 2 of the 24 searches in WT mice (p<0.05, Z-test; *Table 2*). The mean Z-score of all searches in Tau mice was significantly higher than that in WT mice (*Figure 5B*; WT: −0.1 ± 0.3; Tau: 1.4 ± 0.4; p=0.001). Therefore, Tau sequences, but not WT sequences, seen on the track trajectories also appeared robustly during the open box sessions.

Although the animals were allowed to move around freely in the open box, it is possible that some mice ran certain paths repeatedly and resulted in repeatedly occurring firing sequences. To examine this possibility, we plotted the animal's paths between the start and end times of all detected Tau sequences in a search. These paths appeared to be randomly distributed in the box (*Figure 5C*), suggesting that the detected sequences were not linked to particular locations or a particular path. To quantitatively demonstrate this, we drew a vector from the start to the end location of each path. We then plotted all the vectors resulted from a search with their starting positions aligned at the origin (*Figure 5D*). If the paths were not randomly distributed, the vectors should show a preferred direction. However, the vectors of a typical search were randomly orientated (*Figure 5D*). For each search in Tau mice where a significant number of sequences were found, we computed a Rayleigh p value, which tested a null hypothesis that the vectors had no preferred direction. None of the p values for all the 10 searches was significant (*Figure 5E*, p≥0.07), indicating that the null hypothesis could not be rejected. Therefore, the detected Tau sequences in open box sessions were not due to repetitive running behavior. Together with the low location-specificity of Tau neurons in the open box sessions (*Figure 4*), we concluded that Tau sequences seen in the open box were not primarily driven by spatial locations. Even further, these Tau sequences seen in the open box were the same as those during the track running. Therefore, Tau sequences did not distinguish two completely different spaces during two very different behaviors. This is a striking result given the well-known phenomenon of 'remapping' of place cell activities between different environments (*Muller and Kubie, 1987*; *Leutgeb et al., 2005*; *Colgin et al., 2008*).

## Robust firing sequences of CA1 neurons in Tau mice also appeared in novel environments

We have demonstrated that Tau neurons displayed robust firing sequences on a familiar track and in a familiar open box despite their apparent lack of spatial specificity. To understand whether this peculiar feature of Tau neurons requires spatial experience, we also recorded CA1 neurons while mice explored a novel open box for one to two sessions and then ran back and forth (two trajectories) on a novel track for one to two sessions in a novel room. The findings were similar as those seen in the familiar environments.

On the novel track (*Figure 6A*), 79 WT neurons and 92 tau neurons were active on at least one of its trajectories. Their median firing rates during the novel track running sessions were not significantly different (WT: 1.4 [0.4 2.7] Hz, Tau: 1.1 [0.4 3.3] Hz, p=0.71). Whereas the firing locations of WT neurons became stabilized quickly on the novel trajectories, those of Tau neurons stayed unstable from lap to lap (*Figure 6B*). As a result, the median trajectory SI of Tau neurons was much lower than that of WT neurons (WT 0.69 [0.23 1.5] bits/spike, Tau 0.28 [0.086 0.68] bits/spike, p=5.9 ×10$^{-31}$), despite a similar median lap SI (WT 1.2 [0.58 1.8] bits/spike, Tau 1.2 [0.46 1.6] bits/spike, p=0.62). The rate-stability of Tau neurons was much lower than that of WT neurons (WT 0.47 [0.17 0.85], Tau 0.10 [0.03 0.28], p=1.6 × 10$^{-51}$). In the novel open box (*Figure 6C*), 74 WT neurons and 61 Tau neurons were active in at least one session. The median firing rates of these WT and Tau neurons during the novel box sessions were not significantly different (WT: 1.1 [0.4 2.9] Hz, Tau: 1.0 [0.5 2.6] Hz, p=0.38). The rate maps of Tau neurons showed much broader firing areas than those of WT neurons (*Figure 6D*), suggesting a reduced location-specificity in Tau neurons. This was confirmed by the result that the median open SI of Tau neurons was significantly lower than that of WT neurons (WT: 0.28 [0.06 0.71] Hz, Tau: 0.08 [0.02 0.32] Hz, p=1.3 × 10$^{-12}$). These results show that Tau neurons fired with little location-specificity either on the novel trajectories or in the novel open box.

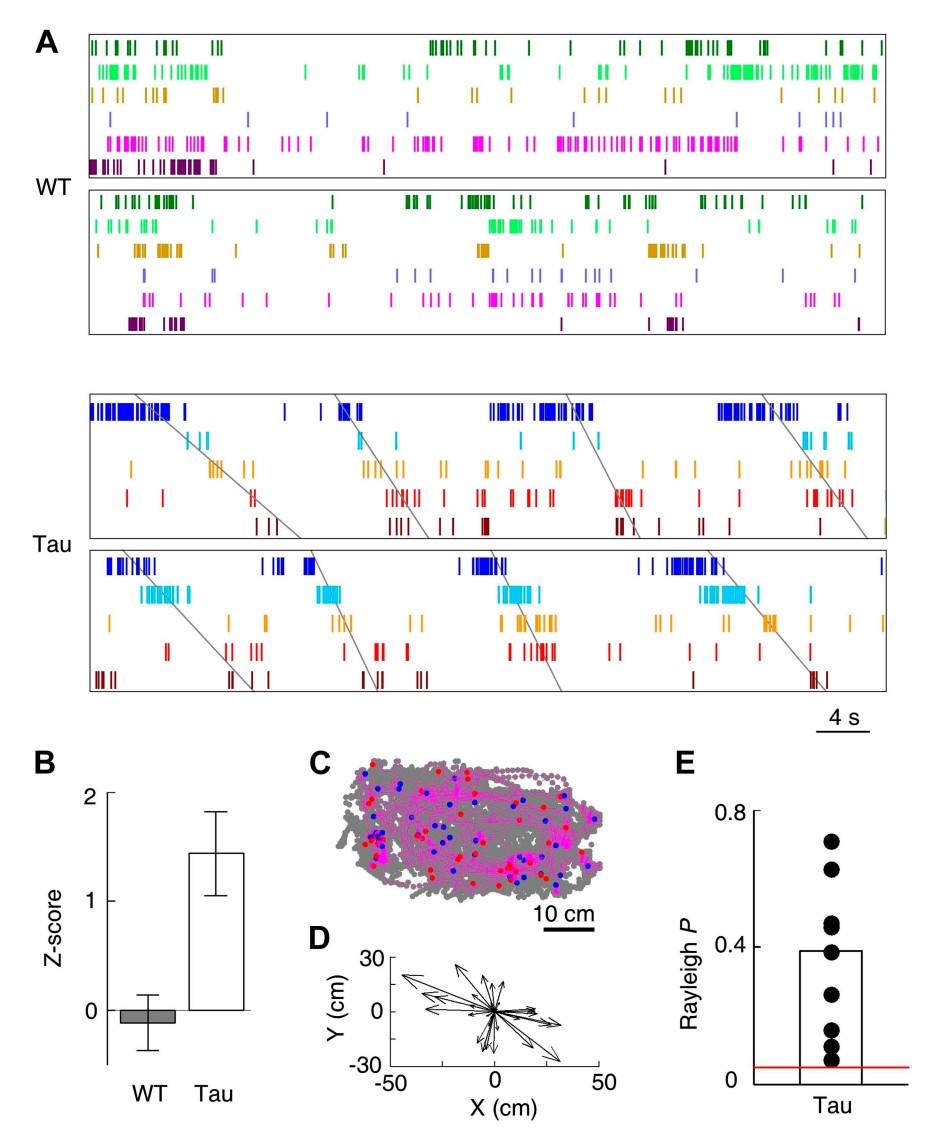

**Figure 5**. Tau, but not WT, sequences seen on the familiar trajectories also appeared in the familiar open box.
(**A**) Spike patterns of the same WT and Tau neurons as in *Figure 3A* within two time periods of an open box
session. Neurons are color-coded. Each tick represents a spike. *Angled lines*: sequences detected as matches
with the Tau sequences shown in *Figure 3A*. See more data from more animals in *Figure 5—figure supple-
ment 1*. (**B**) Mean number (in Z-score) of detected WT and Tau sequences in open box sessions. (**C**) Running
paths of a Tau mouse for all the detected sequences in one open box session. For every sequence, the
animal's position at its start/end time was marked by a *blue/red* dot and its running path in between by a
purple line. *Gray* dots: positions of the animal during the entire session. Note that the sequences occurred
everywhere in the box. (**D**) Vector representation of the paths in panel (**C**). For each path, a vector was drawn
from its start to end location and was plotted with the start location translated to the origin. Note that there
was no preferred direction for the vectors. (**E**) Rayleigh p values for individual searches in open box sessions
of Tau mice. Each dot represents a search for one trajectory template sequence in one open box session. *Bar*:
mean values. *Red line*: p=0.05.

The following figure supplements are available for figure 5:

**Figure supplement 1**. More examples of WT and Tau firing patterns in the familiar open box.

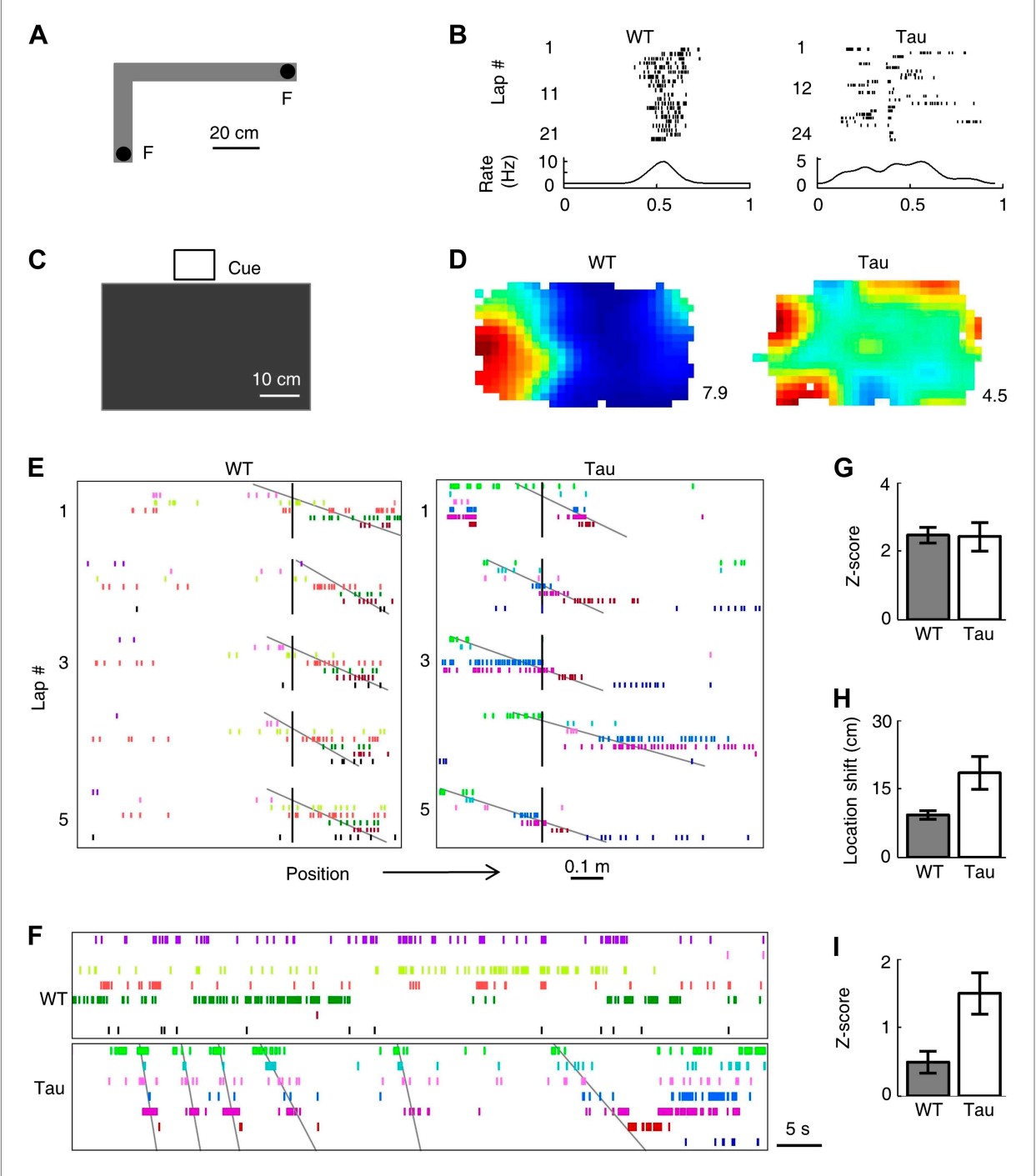

Figure 6. Tau neurons formed robust firing sequences on a novel track and in a novel open box, despite their low location-specificity. (A) The L-shaped novel track. *F*: food wells. (B) Lap-by-lap spike raster of a WT and a Tau neuron on one trajectory of the track (see *Figure 2* for details). (C) The novel open box with its interior color and cue card (*Cue*) shown. (D) Firing rate maps of a WT and a Tau neuron on the open box. *Numbers*: peak (red/black) rates in Hertz. (E) Examples of WT and Tau sequences (see *Figure 3* for details) on one trajectory of the L-track. (F) Spike patterns of the same WT and Tau neurons within a time period of an open box session. (G and H) Mean number of sequences (in *Z*-score) (G) and mean location shift (H) for the sequences detected on the novel trajectories in WT and Tau mice. (I) Mean number of sequences (in *Z*-score) detected during the novel open box sessions in WT and Tau mice.

Nevertheless, Tau neurons, like WT neurons, fired with robust sequences on the novel trajectories. But unlike WT neuron sequences, Tau sequences on the novel trajectories were not anchored to specific locations (*Figure 6E*) and same sequences also appeared in the novel open box sessions (*Figure 6F*). We derived 15 high-order template sequences on the novel trajectories from WT mice and 8 from Tau mice (*Table 3*). The mean *Z*-scores of WT and Tau template sequences on the novel trajectories were similarly high (*Figure 6G*, WT: 2.5 ± 0.2, Tau: 2.4 ± 0.4, p=0.91), but the mean location shift of Tau sequences among laps was greater than that of WT ones (*Figure 6H*; WT: 9.2 ± 0.9 cm, Tau: 18.5 ± 3.6 cm, p=0.0038). The mean location shift values of both WT and Tau sequences shown here were smaller than those on the familiar track, probably due to the shorter length of the novel track. Finally, Tau template sequences were detected in the novel open box sessions significantly more often than WT templates (*Figure 6I*, mean *Z*-scores: WT: 0.5 ± 0.2, N = 30 searches; Tau: 1.5 ± 0.3, N = 12; p=0.0029). Here we emphasize that the novel open box sessions preceded the novel track running. Therefore, Tau sequences seen on the novel trajectories were already preexistent in the prior open box sessions. These findings suggest that the formation of Tau sequences did not primarily depend on spatial experience in an environment.

## External space contributed to the firing activities of CA1 neurons in Tau mice

Does the lack of location-specificity and experience-dependence mean that external space made no contribution to Tau neurons' firing activities at all? To answer this, we first examined whether the firing locations of Tau neurons were completely random. Visual inspection of the lap-by-lap

**Table 3.** High-order sequence analysis results for individual template sequences derived on the novel trajectories

| Template sequence | Animal name | Genotype | Self traj. | Z (S1T1) | Z (S1T2) | Z (S2T1) | Z (S2T2) | Z (O1) | Z (O2) |
|---|---|---|---|---|---|---|---|---|---|
| IHFGE | AN18 | WT | S1T1 | **3.4** | 0.1 | 1.4 | 0.2 | 1.0 | 1.1 |
| HEIFG | AN18 | WT | S1T2 | 0.4 | **4.3** | −1.4 | 2.7 | 1.6 | −0.8 |
| HICFBG | AN18 | WT | S2T2 | 1.9 | 3.5 | −0.3 | **3.6** | 2.5 | 0.8 |
| NAOJBD | AN19 | WT | S1T1 | **2.5** | −1.5 | 1.4 | 0.1 | 0.1 | −0.1 |
| OKDHFBAJN | AN19 | WT | S1T2 | −0.1 | **1.6** | −1.0 | 0.7 | 1.1 | 0.1 |
| LANOB | AN19 | WT | S2T1 | 3.1 | −0.6 | **3.0** | −1.1 | −0.5 | −0.4 |
| OKFCNA | AN19 | WT | S2T2 | −0.4 | 1.3 | −0.4 | **1.7** | 1.6 | 0.1 |
| ISRPBGJXKQAY | AN20 | WT | S1T1 | **1.5** | −1.6 | 2.0 | 1.2 | 0.9 | −1.1 |
| BIPRS | AN20 | WT | S1T2 | −0.1 | **1.1** | 0.2 | −0.3 | 0.3 | 0.2 |
| CTDOISREBXG | AN20 | WT | S2T1 | 1.8 | −0.5 | **1.8** | −0.5 | 1.1 | 0.6 |
| SRUPNL | AN20 | WT | S2T2 | −0.5 | 1.8 | 1.0 | **2.8** | 0.2 | 0.9 |
| IXPHNSC | AN21 | WT | S1T1 | **2.5** | −1.5 | 3.2 | 0.0 | −0.7 | −1.1 |
| PXHYT | AN21 | WT | S1T2 | 0.3 | **1.8** | −0.4 | −0.1 | 1.5 | −0.1 |
| KIJXSC | AN21 | WT | S2T1 | 2.8 | −0.5 | **2.7** | 0.4 | 0.4 | 1.1 |
| UMNSEO | AN21 | WT | S2T2 | −1.6 | −0.4 | 1.1 | **2.5** | 1.7 | 0.7 |
| HGPOL | AN14 | Tau | S1T1 | **1.7** | −0.1 | 1.1 | −1.4 | 1.4 | −0.3 |
| GEDMHCB | AN15 | Tau | S1T1 | **3.3** | 1.7 | 2.8 | 1.6 | 0.5 | 3.1 |
| DMHCB | AN15 | Tau | S2T1 | 4.2 | 1.9 | **3.8** | −0.4 | 0.8 | 0.7 |
| GEDM | AN15 | Tau | S2T2 | 3.3 | 2.5 | 2.9 | **2.1** | 1.4 | 3.1 |
| ABDE | AN24 | Tau | S1T1 | **2.6** | 3.0 | NA | NA | 1.3 | NA |
| DEABI | AN24 | Tau | S1T2 | 4.6 | **3.5** | NA | NA | 2.6 | NA |
| SGHER | AN25 | Tau | S1T1 | **2.3** | 1.4 | NA | NA | 2.0 | NA |
| KMFGHE | AN25 | Tau | S1T2 | 1.0 | **2.3** | NA | NA | 1.4 | NA |

See *Table 2* for details.

spike trains of Tau neurons revealed that the firing locations of Tau neurons were often consistent among a few laps on both the familiar (*Figure 2B*) and novel (*Figure 6B*) trajectories, suggesting that the locations were not completely random. Indeed, randomly sliding rate curves of individual laps further significantly reduced the median rate-stability of Tau neurons on both familiar (−0.0020 [−0.034 0.042], p=9.3 × 10$^{-49}$ compared with actual rate-stability) and novel (−0.0020 [−0.032 0.049], p=1.8 × 10$^{-60}$) trajectories, indicating that spatial locations still modulated Tau neurons' firing activities.

Second, we examined whether Tau neuron sequences were trajectory-selective. On both the familiar and novel tracks, most animals ran two different trajectories (back and forth between two food wells) in each of the two running sessions. For each template sequence derived on a trajectory in a session, we counted the occurrence of the sequence (in *Z*-scores) not only on the trajectory itself (Self), but also on the same trajectory (ST) in the other session and on the other trajectory (OT) in the same session. On the familiar track, the mean *Z*-scores of Self and ST were similar for both WT (Self 2.5 ± 0.3, *N* = 13 templates; ST 2.3 ± 0.2, *N* = 13; p=0.53) and Tau (Self 2.3 ± 0.3, *N* = 8; ST 1.6 ± 0.5, *N* = 8; p=0.27) sequences, but there was a significant difference between Self and OT for both WT (OT 0.29 ± 0.3, *N* = 13; p=5.8 × 10$^{-6}$) and Tau (OT 0.10 ± 0.2, *N* = 8; p=6.1 × 10$^{-5}$) sequences (*Figure 7A*). The result shows that Tau sequences, like WT sequences, occurred consistently on same trajectories across different sessions and distinguished between different trajectories within same sessions. On the novel track, whereas the mean *Z*-scores of Self and ST remain comparable for both WT (Self 2.5 ± 0.2, *N* = 15; ST 1.7 ± 0.3, *N* = 15; p=0.06) and Tau (Self 2.4 ± 0.4, *N* = 8; ST 2.6 ± 0.6, *N* = 4; p=0.79) sequences, there was a significant difference between Self and OT only for WT (OT −0.26 ± 0.2, *N* = 15; p=3.0 × 10$^{-9}$), but not for Tau (OT 1.9 ± 0.5, *N* = 8; p=0.51) sequences (*Figure 7B*), suggesting that Tau sequences, unlike WT sequences, did not distinguish the two trajectories on the novel track. This analysis shows that Tau sequences were originally not trajectory-selective on the novel track, but became so on the familiar track. Therefore, although Tau neuron's firing activities were not location-specific and Tau sequences were not primarily driven by space, there was still a modulation of Tau neuron's firing activities by spatial experience and spatial trajectory.

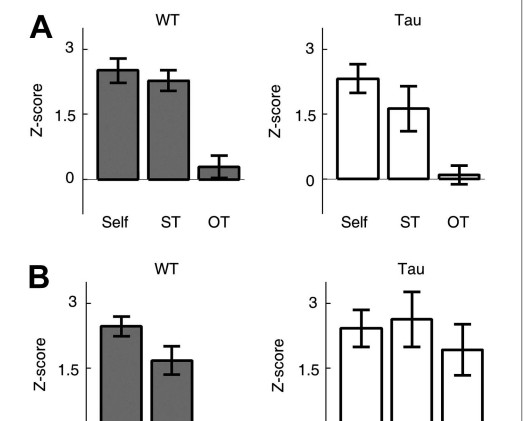

**Figure 7**. Tau sequences were trajectory-selective on the familiar track, but not on the novel track. (**A** and **B**) Mean number of sequences on three types of trajectories (*Self*, *ST*, *OT*) of the familiar (**A**) and novel (**B**) track. For each template sequence derived on a trajectory of a session, we computed the number of sequences (in *Z*-score) that matched with the template on the trajectory itself (*Self*), on the same trajectory (*ST*) but in the other session, and on the other trajectory (*OT*) in the same session.

## There was no apparent spatial or temporal periodicity in the occurrence of Tau sequences

Since same Tau sequences sometimes occurred multiple times in a single lap (*Figure 3A*), we asked whether they tended to appear periodically in space or time. For all the sequences matched with a template in a session, we visualized the spatial/temporal gaps between neighboring sequences by a raster plot and computed the spatial/temporal auto-correlogram of the sequences using their locations/times to examine the periodicity (*Figure 8*). Neighboring WT sequences in track sessions displayed a consistent spatial gap and there were periodic peaks in their spatial auto-correlograms (*Figure 8A*, *left*). The spatial period was the same as the track length (~2 m for the familiar track, ~1 m for the novel track), which was expected, given the repetitively running of same track trajectories. The periodicity was much less obvious in the temporal domain (*Figure 8A*, *right*), because the amount of time that animals spent running and at the food sites varied from lap to lap. In contrast, Tau sequences during track sessions did not show any consistent spatial or temporal gaps in their raster plots and no obvious, periodic peaks were observed in either the spatial

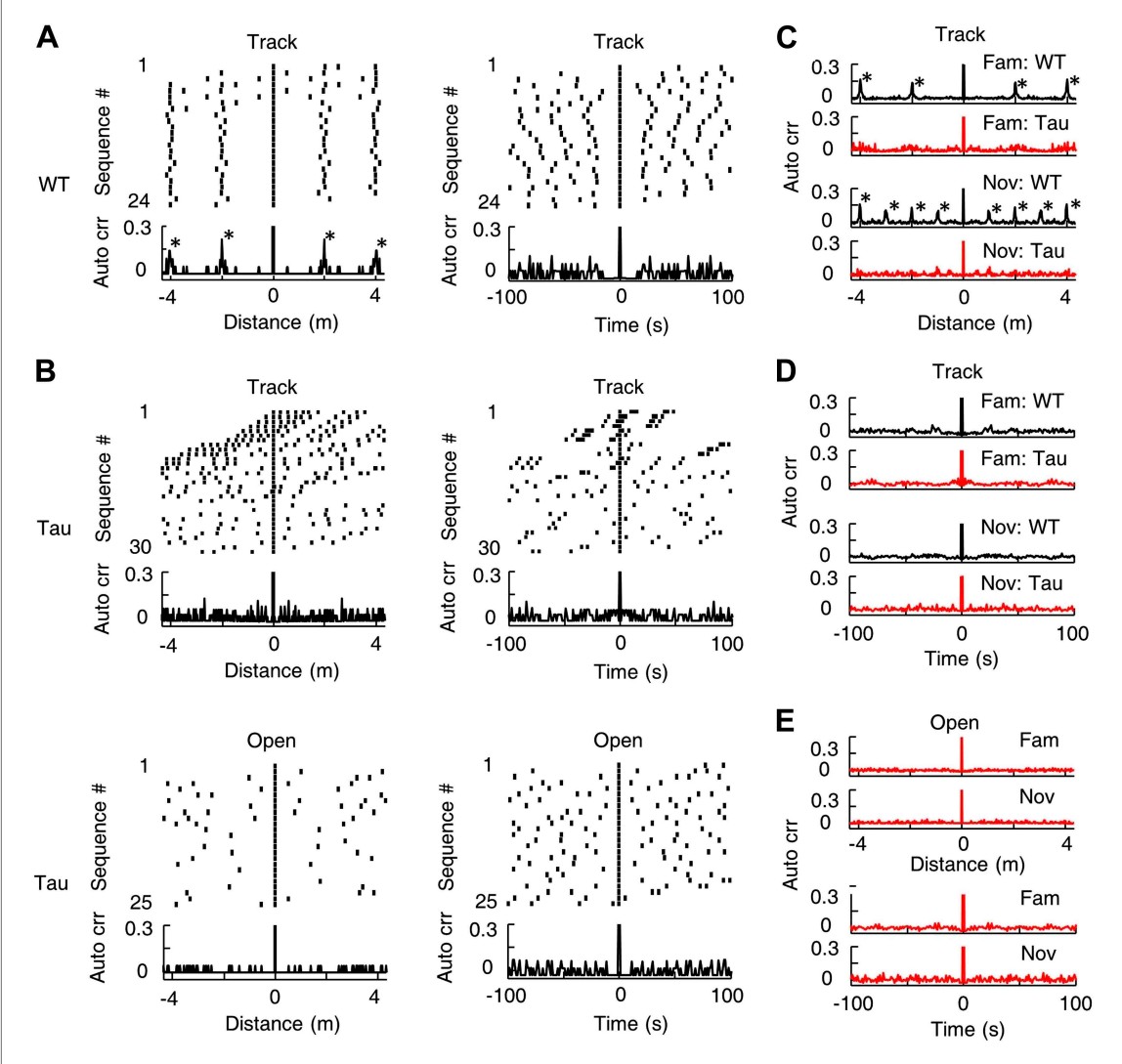

**Figure 8**. Tau sequences did not display obvious periodicity in space or time. (**A**) The raster and auto-correlogram (*Auto crr*) of all sequences matched with a template during a track session of a WT mouse, plotted in the spatial (*left*) and temporal (*right*) domain. Each tick represents a sequence. For each sequence, we aligned its location/time at 0 and plotted the sequences occurring before and after at corresponding location/time gaps (*top*). The histogram of this raster plot is equivalent to the auto-correlogram of the sequence locations/times (*bottom*). The auto correlation value at 0 was truncated. Note the regular intervals in the spatial raster plot and periodic peaks in the spatial auto-correlogram. (**B**) Same as (**A**), but for the sequences matched with a template during a track session (*Track*) and an open box session (*Open*) of a Tau mouse. Note the inconsistent spatial intervals and the disappearance of the peaks in the track spatial auto-correlogram. (**C**) The average sequence spatial auto-correlograms during familiar (*Fam*; WT: *N* = 9 templates; Tau: *N* = 6) and novel (*Nov*; WT: *N* = 12 templates; Tau: *N* = 7) track sessions of WT and Tau mice. (**D**) Same as (**C**), but in the temporal domain. (**E**) The average sequence auto-correlograms of Tau mice in familiar (*Fam*; *N* = 10 searches) and novel (*Nov*; *N* = 4) open box sessions in the spatial (*top*) and temporal domain (*bottom*).

or temporal auto-correlograms (*Figure 8B*). The spatial periodicity at the track length disappeared apparently because Tau sequences were not anchored to track locations. The average auto-correlogram across all sessions confirmed this observation on both the familiar and novel tracks (*Figure 8C,D*). Using the same analysis, Tau sequences in open box sessions did not show apparent spatial or temporal periodicity either (*Figure 8B,E*). Therefore, there was no strong evidence for a periodic occurrence of Tau sequences.

## The robust firing sequences in Tau mice could not be explained by simple behavioral parameters

We have shown the existence of robust Tau sequences that were not anchored to specific locations and even did not distinguish different environments. Is it possible that these sequences were

produced simply by some peculiar behaviors of Tau mice? Track running and free exploration of open boxes are simple behavioral tasks that both WT and Tau mice were able to perform. *Figure 9A,B* show the accumulative running paths and linearized individual laps of a WT and Tau mouse on the familiar rectangular track. *Figure 9C* shows the accumulative running paths of the same mice in the familiar open box. As seen in these raw plots, the overall behavior of the WT and Tau mouse was similar. We then quantified the behavior using the following parameters. The first, most straightforward parameter is the running speed during the running of a track trajectory or during an open box session. In addition, for each trajectory of a track, we measured the animal's performance by the number of laps on the trajectory and quantified the quality of the running by computing the cumulative travel distance per lap, the number of stops (defined as speed <4 cm/s lasting ≥2 s) per lap, and the stopping duration per lap. These parameters were computed for 54 familiar (WT: 22, Tau: 32) and 46 novel (WT: 20, Tau: 26) trajectories and 27 familiar (WT: 11, Tau: 16) and 25 novel (WT: 12, Tau: 13) open box sessions. The results are shown in *Figure 9D–H*. On familiar trajectories, Tau mice completed less number of laps and ran with a slower speed than WT mice, but their running quality as measured by stopping and travel distance was similar. On novel trajectories, Tau mice behaved similarly as WT mice or even better on one measure (less number of stops per lap). In open boxes, Tau mice in general ran slightly faster than WT mice, but the difference was significant only in the novel open box. Since robust Tau sequences were found on both the familiar and novel tracks and these sequences appeared in both the familiar and novel open boxes, it is unlikely that the running behavior per se was responsible for the differences between WT and Tau sequences.

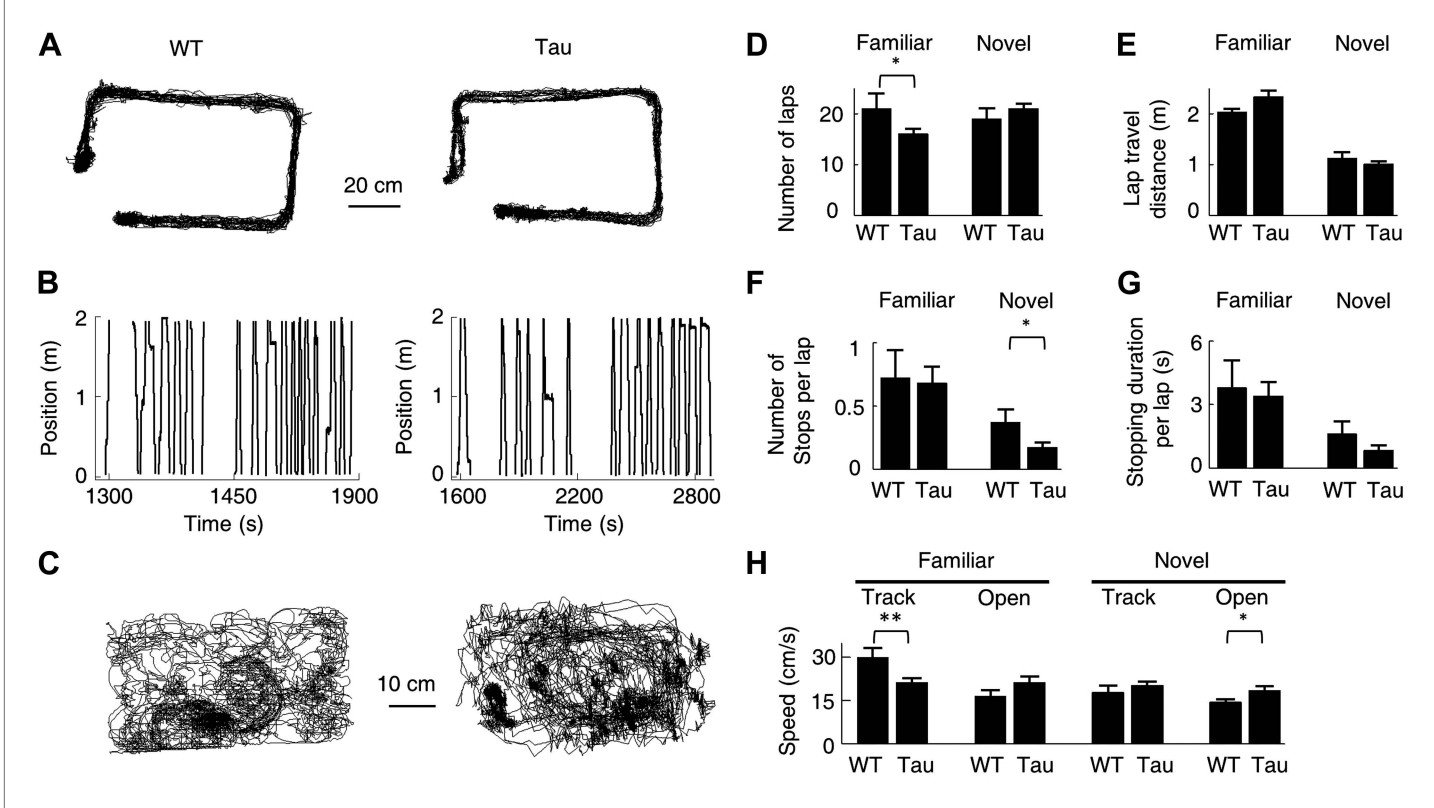

**Figure 9**. Behavioral quantifications of WT and Tau mice. (**A**) The accumulative running paths of a WT and Tau mouse on the familiar rectangular track. (**B**) The paths in (**A**) were linearized lap-by-lap on the two track trajectories (ascending and descending lines) and plotted against time. The positions at the food wells were truncated. (**C**) The accumulative running paths of the WT and Tau mouse during an open box session. (**D–G**) the mean number of laps per session (**D**), mean travel distance per lap (**E**), mean number of stops per lap (**F**), and mean stopping duration per lap (**G**) for WT and Tau mice during familiar and novel track sessions. (**H**) Mean running speed of WT and Tau mice during familiar and novel track (*Track*) and open box (*Open*) sessions. *p<0.05; **p<0.01.

## Theta oscillations were qualitatively similar between WT and Tau mice

Theta (6–12 Hz) oscillation in the CA1 local field potentials (LFPs) is associated with normal place cell activities (*Buzsaki, 2002*) and is known to organize multiple place cells into firing sequences at theta time scale (*O'Keefe and Recce, 1993*; *Skaggs et al., 1996*; *Dragoi and Buzsaki, 2006*). We next asked whether theta oscillations were altered in Tau mice. As seen in *Figure 10A*, the raw and filtered LFP traces of a Tau mouse appeared similar to those of a WT mouse, except that the overall magnitude of the Tau mouse's LFPs was reduced, which is expected because of the massive neurodegeneration. Power spectral density (PSD) analysis of the raw traces showed a clear peak within the theta band in the LFP PSDs of both mice, but with a reduced amplitude in the Tau mouse's (*Figure 10B*, *left*). However, when the PSD values were normalized by the total LFP power, the peak of the Tau mouse's PSD was comparable with that of the WT mouse (*Figure 10B*, *right*), indicating the presence of prominent theta oscillations in the LFPs of the Tau mouse.

We quantified the peak frequency and the absolute and normalized theta power for each LFP trace of a recording session. These parameters were compared among WT, Tau, and mice of two other control genotypes (+/−, −/+, see 'Materials and methods'), which showed no obvious tau-mediated neurodegeneration in CA1 (*Figure 1* and *Table 1*). There was a slight reduction in theta peak frequency in Tau mice (~8.5 Hz, compared with ~9.5 Hz in control mice) during familiar track and open box sessions. This difference became even less obvious in novel sessions (*Figure 10C*). The absolute theta power was apparently reduced in Tau mice from those of control animals (*Figure 10D*). However, the normalized theta power in Tau mice was significantly lower than that in control mice only during familiar track sessions, but not during open box sessions or novel track sessions (*Figure 10E*). We also restricted this analysis on LFPs within the start and end times of detected sequences in WT and Tau mice. The observation was similar (*Figure 10F–H*). Therefore, although LFPs in Tau mice were overall smaller, prominent theta oscillations were still present and qualitatively similar to those of WT mice, suggesting that abnormal theta oscillation is unlikely the cause of the difference between Tau and WT sequences.

## There was a small gender difference in Tau mice

We also considered additional factors that may affect Tau neuron firing activities. First, we analyzed whether there was a gender difference in Tau mice. Rather than the sequence-related parameters, we used the rate-stability on track trajectories for this purpose because it allowed the comparison between male and female mice with sufficient samples (*Figure 11A,B*). Between those neurons recorded from male and those from female WT mice, there was no significant difference in the mean rate-stability either on familiar (male 0.70 [0.33 0.88], female 0.65 [0.29 0.86], p=0.21) or novel (male 0.54 [0.19 0.83], 0.46 [0.16 0.86], p=0.22) trajectories. Between those from male and female Tau mice, the mean rate-stability was similar on novel trajectories (male 0.10 [0.023 0.26], female 0.10 [0.037 0.31], p=0.51). But on the familiar trajectories, the mean rate-stability of the neurons from female Tau mice was significantly greater than that of male Tau mice (male 0.047 [−0.0035 0.14], female 0.13 [0.031 0.42], p=$5.5 \times 10^{-16}$). For both genders and on both the familiar and novel trajectories, the mean rate-stability of WT neurons was always much greater than that of Tau neurons (p≤$1.3 \times 10^{-14}$). These results indicate that the firing stability of CA1 neurons was severely compromised in Tau mice of both genders, but CA1 neurons in female Tau mice did better on familiar trajectories than those in male Tau mice.

Second, we analyzed an additional 163 neurons recorded from four mice with two other control genotypes (+/−, −/+). We found that they were much more stable than Tau neurons and largely similar to WT neurons. On familiar trajectories (*Figure 11C*), the median rate-stability values of both +/− and −/+ neurons were not different from that of WT ones (+/−: 0.62 [0.35 0.80]; −/+: 0.63 [0.39 0.85]; p≥0.11 compared with WT). Both were significantly greater than that of Tau neurons (p≤$1.5 \times 10^{-30}$). On novel trajectories (*Figure 11D*), the median rate-stability of −/+ neurons was not different from, but that of +/− was slightly smaller than, that of WT neurons (+/− 0.41 [0.090 0.69], −/+ 0.49 [0.21 0.76], p=0.016 between +/− and WT, p=0.90 between −/+ and WT). Both were significantly greater than the median rate-stability of Tau neurons (p≥$1.4 \times 10^{-21}$).

## Discussion

We have examined the HP spatial memory code in a tauopathy mouse model, the transgenic rTg4510 mice. Our data show that CA1 neurons in the transgenic mice do not fire at specific locations, but still form robust firing sequences. These sequences are not anchored to precise locations of spatial

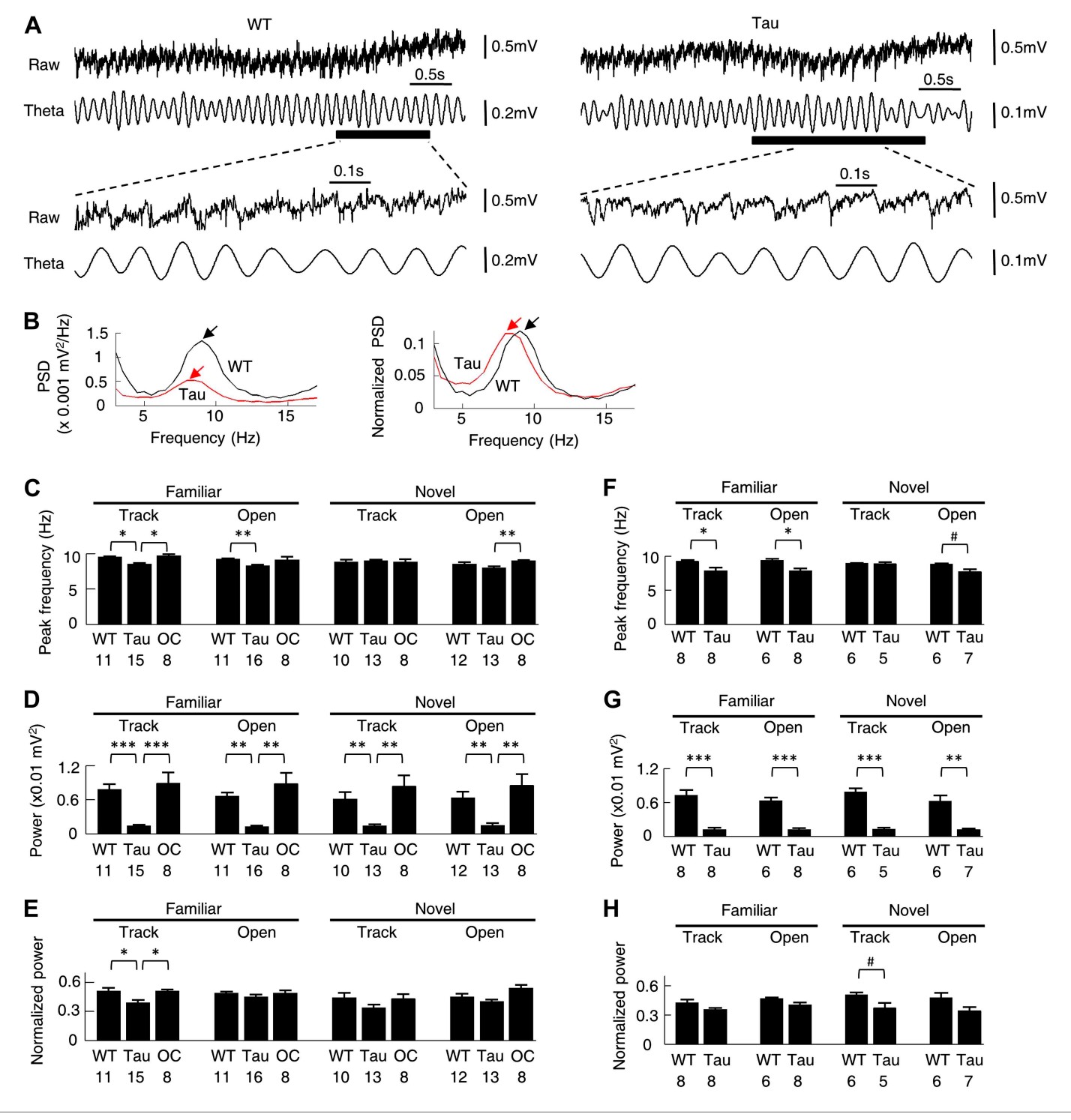

**Figure 10**. Theta oscillations were present in Tau mice. (**A**) Raw and theta-filtered (6–12 Hz) LFP traces recorded from a WT and Tau mouse. *Black bars*: time windows during which a WT and a Tau firing sequence were detected. The raw and theta-filtered LFP traces within the time windows are expanded on the bottom. Note the similar theta oscillations between the WT and Tau mouse, but with different scale bars. (**B**) The absolute (*left*) and normalized (*right*) PSD curves of the raw LFP traces shown in (**A**). *Black*: WT; *Red*: Tau. *Arrows*: peaks in the theta frequency band. (**C**–**E**) mean peak theta frequency (**C**), mean absolute theta power (**D**), and mean normalized theta power (**E**) of raw LFPs recorded from WT and Tau mice and from mice with two other control genotypes (OC; +/– and –/+ combined), during familiar and novel track running (*Track*) and open box sessions (*Open*). *Numbers*: number of samples. (**F**–**H**) same as in (**C**)–(**E**), but for raw LFPs within the start and end times of detected WT and Tau firing sequences. #p<0.05; *p<0.01; **p<0.001; ***p<0.0001. For (**C**)–(**E**), the significance threshold was lowered to 0.01 because of the multiple (3) comparisons within a group and therefore only those comparisons with p<0.01 were marked.

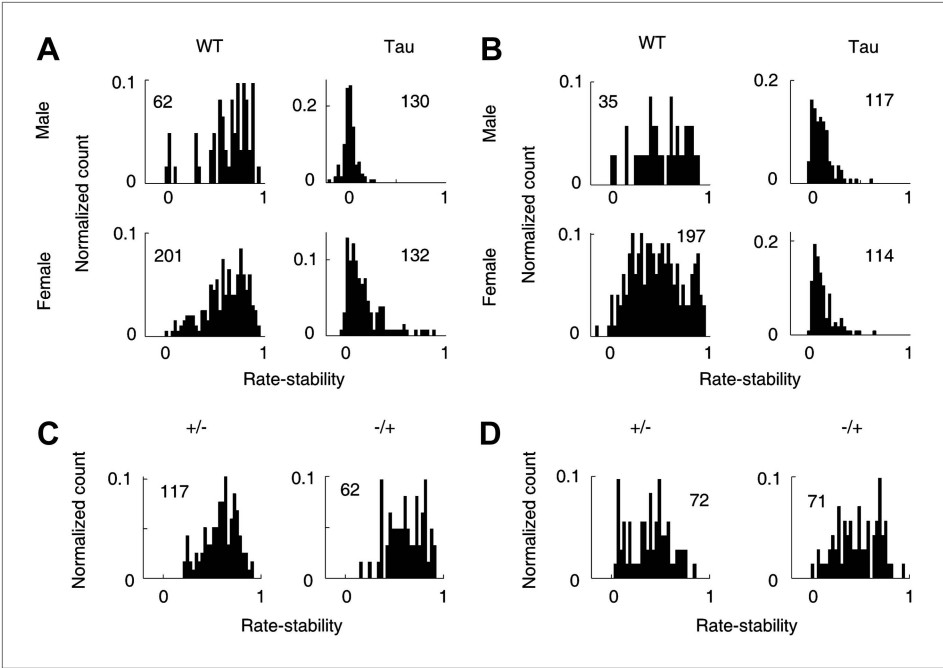

**Figure 11**. Gender and genotype differences. (**A** and **B**) Distributions of rate-stability for WT and Tau neurons recorded from male and female mice on familiar (**A**) and novel (**B**) trajectories. (**C** and **D**) Distributions of rate-stability for neurons recorded from mice with other two control genotype (+/−, −/+) on familiar (**C**) and novel (**D**) trajectories. Plots are histograms normalized by total numbers of samples (*numbers* shown), each computed for one neuron on one trajectory.

trajectories and do not even distinguish between two very different spatial environments. The sequences seen on novel trajectories already exist in a prior session of free exploration in an open box. Therefore, CA1 firing sequences in the transgenic mice become so rigid that they no longer primarily respond to spatial environments or spatial experience, and therefore no longer capable of encoding spatial memories.

The lack of both spatial specificity and experience-dependence indicates that external spatial input in Tau mice is incapable of taking control of Tau neurons. Nevertheless, Tau neurons do not fall into disorganized firing patterns, but fire with robust sequences. If not external space, what could be the driven force that assembles Tau neurons into these sequences? It has been recently shown that internal brain activities, those related to memory recall (*Gelbard-Sagiv et al., 2008*; *Pastalkova et al., 2008*) and those encoding time (*MacDonald et al., 2011*; *Eichenbaum, 2013*), can produce CA1 firing sequences similar to those shown here. Therefore, here we propose that internal activities underlie the rigid firing sequences in the transgenic mice. On the other hand, although the firing activities of CA1 neurons in the transgenic mice were not location-specific, they are still modulated by external space and spatial experience (*Figure 7*). The existence of this external modulation together with the rigid firing sequences paint a picture of 'cued false activation'. When the transgenic mice are placed in a space, instead of forming/retrieving the space's memory code, CA1 neurons are cued to activate those internally driven activity patterns irrelevant to the current space. According to this interpretation, the reason why these mice cannot form new spatial memories is because the HP network is dominated by internal brain activities.

This interpretation implies that there exists a direct competition between external and internal inputs in HP. Therefore, our data provide evidence that abnormal external-internal interaction may contribute to memory deficits in neurodegenerative diseases. Previous studies have shown that CA1 firing sequences can occur during immobility (*Foster and Wilson, 2006*; *Diba and Buzsaki, 2007*; *Davidson et al., 2009*; *Karlsson and Frank, 2009*; *Carr et al., 2011*) or when the animal is spatially restricted (*Pastalkova et al., 2008*; *MacDonald et al., 2011*). These sequences are believed to be internally generated because external spatial input stays stationary in these situations. In our

experiments, the animals' paths between the start and end times of Tau sequences spanned a distance on the tracks (*Figures 3A and 6E*) and in the open boxes (*Figure 5C*), clearly showing that the animals were moving when Tau sequences occurred. The sequences also occurred in accompany with prominent theta oscillations (*Figure 10A,H*). Therefore, the non-spatially driven firing sequences were found even when animals were actively moving through space. In this case, the external spatial input associated with the movement is overtaken by internal input in the control of CA1 neurons' firing activities. This further extends the recent findings that pre-existing neuronal sequences may influence future memory codes (*Gupta et al., 2010*; *Dragoi and Tonegawa, 2011*).

What changes in the neural circuits of Tau mice could produce the abnormal completion between internal and external input? Internal activities could originate in CA3, an area where neurons can drive each other via its extensive recurrent connections (*Lisman, 1999*; *Nakazawa et al., 2002*). External sensory input reaches CA1 likely through the direct pathway via the entorhinal cortex. In Tau mice, there is evidence that tau pathology starts earlier in the cortex including the entorhinal cortex than in HP, and within HP, neuronal loss is more prominent in CA1 than in CA3 (*Ramsden et al., 2005*). It is likely that the external input to CA1 through the entorhinal cortex is weakened by the pathology, but the internal input from CA3 is relatively spared. In the entorhinal cortex, grid cells fire spikes in a spatially periodic fashion (*Hafting et al., 2005*). In our data, although Tau sequences sometimes occurred repeatedly in a running lap, there was no evidence for either spatial or temporal periodicity in Tau sequences (*Figure 8*). Also Tau cells' firing activities were not tied to specific locations on the tracks (*Figure 2B*) and did not show obvious periodic patterns in the open boxes (*Figure 4B*). Therefore, Tau sequences are unlikely resulted from periodic firing of entorhinal grid cells. Other potentially relevant changes in Tau mice could include interneurons and theta oscillations, since they greatly shape the firing activities of HP place cells (*O'Keefe and Recce, 1993*; *Skaggs et al., 1996*; *Buzsaki, 2002*; *Dragoi and Buzsaki, 2006*). In Tau mice, the CamKII gene that induces tau pathology and neurodegeneration is not expressed in interneurons and thus should mainly affect pyramidal neurons in HP (*Mayford et al., 1996*; *Sik et al., 1998*; *Ramsden et al., 2005*). However, interneurons could still die or alter their properties as adaptive changes to the tau pathology and loss of pyramidal neurons. In our datasets, we only recorded 11 putative interneurons from WT mice and 12 from Tau mice. There was a trend that Tau interneurons fired with a lower rate than WT ones (WT: 24.2 [11.7 46.7] Hz, Tau: 14.8 [8.8 27.2] Hz, p=0.05). This rate change could be one of the adaptive changes in Tau mice to keep the overall activity of remaining HP pyramidal neurons at normal level. Regarding theta oscillations, although the medium septum, the structure important for theta generation in HP, could be affected by tau pathology, based on the known expression pattern of CamKII (*Odeh et al., 2011*), our data show that prominent theta oscillations are still present in the CA1 LFPs of Tau mice. In particular, the proportion of theta power in the total LFP power is in many cases similar between WT and Tau mice (*Figure 10E,H*). Therefore, theta oscillation unlikely plays a major role in the generation of abnormal Tau sequences. Based on these considerations, we propose that the internal input from CA3, by winning over the external input via the direct pathway from the entorhinal cortex, is the main driving force for producing the rigid Tau sequences. As such, Tau sequences mainly reflect the internal information such as old memories stored in the CA3 recurrent connections.

Our data reveal a subtle but significant difference between male and female Tau mice: The firing stability of CA1 neurons on familiar trajectories in female is greater than that in male mice. This is somewhat unexpected, given a previous report that female tau mice show higher level of abnormally phosphorylated tau and worse performance in the Morris water maze task (MWM) than male mice (*Yue et al., 2011*). Other previous studies on another tau-related mouse model have found more extensive β-amyloid, but not tau, pathology (*Hirata-Fukae et al., 2008*) and more severe impairments on MWM in females than in males (*Clinton et al., 2007*). One possible explanation for the discrepancy is the age of animals used in our study. The sex difference of Tau mice was previously examined at 5.5 months old (*Yue et al., 2011*). At much older ages (7–9 months) as in our study, tau pathology becomes much more extensive and possibly saturated and thus the sex difference may become less significant (*Ramsden et al., 2005*). Another possibility is that tau pathology may not only affect spatial memory representation, but also other factors important for MWM. For example, it has been proposed that behavioral testing produces more stress in females than males and thus greater behavioral deficits under pathological conditions (*Clinton et al., 2007*). In our experiments, the animals had been handled for at least 1 month (for tetrode adjustment and maze training) before recordings began. The mice in our study might have a reduced level of stress, which exposed the subtle advantage of females in the

firing stability of CA1 cells. This advantage in females could be compensated by a stronger stress response to MWM, producing an overall greater impairment on task performance.

What pathological features in the transgenic mice are responsible for the observed rigid firing sequences? Our recordings were conducted on the transgenic mice at 7–9 month old, which is a late, mature stage of tau pathology that displays all typical pathological features including hyperphosphorylated tau, tau tangles, and neurodegeneration (*Ramsden et al., 2005*; *Santacruz et al., 2005*). Previous experiments show that stopping the production of the transgenic tau is sufficient to halt the memory deficits at the behavioral level (*Ramsden et al., 2005*; *Santacruz et al., 2005*), pointing to a role of the hyperphosphorylated tau protein itself. Another possibility is that neurodegeneration reduces the neuron number in HP and cortex and result in a limited capacity for memory storage. Future studies on tauopathy models at various pathological stages may uncover the exact features of tau pathology and/or neurodegeneration that cause the observed abnormalities. Nevertheless, the current study allows us to identify the functional alterations of CA1 neurons at a mature stage of tau pathology and lay the foundation for probing their pathological causes.

Currently, no animal models can fully recapitulate the pathological changes in human AD patients (*Ashe and Zahs, 2010*). The limitation in the animal model calls for caution in applying the finding to the human disease. However, tau pathology is a key feature of AD (*Wenk, 2003*; *Ashe and Zahs, 2010*) and plays a key role in its behavioral phenotypes (*Ashe and Zahs, 2010*). This model does mimic key features of human tauopathy including its age-dependent progression (*Ramsden et al., 2005*; *Santacruz et al., 2005*). The functional changes at the cellular level in HP identified in this animal model, such as the low location-specificity and rigid firing sequences, possibly also occur in the human brain with similar tau pathological features. Interestingly, our interpretation that internally generated activities obstruct the formation of new memories is consistent with two typical memory symptoms seen in AD patients: the inability to form new memories (*Carlesimo and Oscar-Berman, 1992*; *Salmon and Bondi, 2009*) and the intrusion of old memories (*Butters et al., 1987*; *De Anna et al., 2008*). This illustrates that studying the functional changes in memory circuits of animal models can generate novel insight into the memory symptoms of human neurodegenerative diseases including AD.

## Materials and methods

### Animals

The mouse colony was maintained by an activator and a responder mouse lines (*Ramsden et al., 2005*; *Santacruz et al., 2005*). The activator, bred in a 129S6 background strain, carried the tTA transgene under the CaMKIIa promoter. The responder, bred in the FVB/N strain, carried a transgene encoding the human four-repeat tau with the P301L mutation ($hTau_{P301L}$). The F1 offspring of the two mouse lines that carries both the $hTau_{P301L}$ and tTA transgenes (hTau+/tTA+, Tau mice) has been shown to overexpress the human tau in the forebrain and display age-dependent progression of tau pathology and neurodegeneration in the hippocampus (HP) and the cortex (*Ramsden et al., 2005*; *Santacruz et al., 2005*). The animals used in this study were Tau mice, their wildtype (hTau−/tTA−, WT) littermates, and littermates with the other two control genotypes, hTau+/tTA− (+/−) and hTau−/tTA+ (−/+), all at 7–9 month old (*Table 1*). Both male and female animals were used.

### Experimental procedure

A hyperdrive containing eight tetrodes was implanted during a surgery onto the skull of each mouse used in this study. Over the course of 2–4 weeks post surgery, tetrodes were slowly moved down to the CA1 pyramidal cell layer of HP. Starting about one week post surgery, the animal was food-deprived, with weight maintained above 85% of the *ab libitum* level, and trained in a familiar room to run back and forth (two trajectories) on a ~2 m long rectangular track (*Figure 2A*) for food reward (liquid condensed milk) and to freely explore in a 50 × 30 cm open box (*Figure 4A*). The animal was trained for at least 3 weeks and 15–30 min in each apparatus each day.

Recording in the familiar room started when multiple, stable single-units (neurons) had been obtained in CA1 and the animal achieved a performance of at least 10 laps per trajectory on the track. The daily recording procedure consisted of two sessions on the track, followed by one to two sessions in the open box. Each session lasted about 15 min and there was a 15-min break between any two sessions, during which the animal rested on an elevated dish (25 cm high, 18 cm in diameter) inside an enclosed box. The recording was repeated 3–10 days.

After completing the recordings in the familiar room, recording continued in a novel, different room while the animal freely explored a novel open box (30 × 50 cm, *Figure 6C*) and ran back and forth on a novel ~1 m long L-shaped track (*Figure 6A*). The animal had never been in the novel room or in the novel apparatuses before. The recording schedule was one to two open box sessions, followed by one to two sessions on the L track. Each session lasted about 15 min and there was a 15-min break between any two sessions.

After the recording, the animal was sacrificed using pentobarbital overdose (200 mg/kg, IP injection). Electrical current (30 µA, 6–10 s) was passed through the tetrodes to create small lesions at the recording sites. The animal's brain was dissected out and fixed in 10% formalin for 24 hr. After its weight and size were measured, the brain was then sectioned for immunohistochemistry.

## Surgery

The animal was fixed on a stereotaxic device with body temperature maintained at 36–38 °C and anesthetized with continuous flow of 0.5–2% inhalation anesthetic isoflurane. The flow was adjusted to keep the animal's breathing rate at 40–80 per min. Ten anchor screws were mounted onto the skull. An exposure was made at the coordinates anteroposterior −2.0 mm, mediolateral 1.5 mm from Bregma. The hyperdrive cannula containing eight tetrodes was lowered to the exposure just above the brain. The drive was fixed in place by dental acrylic. Analgesic (ketoprofen, 5 mg/kg) was injected subcutaneously before the animal recovered from the anesthesia.

## In vivo electrophysiological recording

Tetrode recording was made using a DigitalLynx acquisition system (Neuralynx, Bozeman, MT) and followed the published procedures (*Ji and Wilson, 2007*). For spike recording, voltage signals from four channels of a tetrode were digitally band-pass filtered between 600 Hz and 9 kHz. Spikes were detected with any of the four channels crossing a pre-set triggering threshold (50–70 µV) and were sampled at 32 kHz. Broad band (0.1 Hz–1 kHz) local field potentials (LFPs) were sampled at 2 kHz sampling rate. Two color diodes (red, green) were mounted over the animal's head to track its positions. Positions were sampled at 33 Hz with a resolution approximately 0.2 cm.

## Cresyl violet staining

Coronal sections with 20–70 µm thickness were stained using 0.2% Cresyl violet and cover-slipped for storage. Tetrodes were identified (*Figure 1*), by matching the lesion sites with tetrode depths and their relative positions.

## Data analysis

Data recorded from a total of 18 mice with four different genotypes were analyzed (*Table 1*). For each animal, one day's data in the familiar room (the day with most neurons recorded) and one day's data in the novel room (the first day of exposure) were included in the analysis. Neurons were manually sorted using xclust (Matthew A Wilson, MIT). In total, 333 CA1 neurons in the familiar room and 327 in the novel room were obtained. Among them, 246 in the familiar and 221 in the novel room were classified as putative pyramidal neurons active either on a trajectory of a track or in an open box session (mean firing rate: ≥0.5 Hz and <7 Hz). 11 in the familiar and 12 in the novel room were classified as putative interneurons (mean rate ≥ 7 Hz). Further analysis was performed only on the active, putative pyramidal neurons unless otherwise specified. Results are presented as median and [10% 90%] range values with significance determined by the non-parametric *ranksum* test, or as mean ± SEM values with significance determined by one tailed *t*-test, unless specified otherwise.

### Rate curves, rate maps, and spatial information (SI)

A one-dimensional (1D) trajectory was linearized and divided into 2 cm spatial bins. Similarly, a 2D open box was divided into 2 × 2 cm grids. The rate curve or rate map of a neuron was obtained by computing the neuron's firing rate at each bin or grid. The rate $x_i$ at the *i*-th bin or *i*-th grid was the number of spikes divided by the total time the animal spent at the bin/grid (occupancy time, $t_i$). Then, SI was (*Skaggs et al., 1993*),

$$SI = \sum_{i=1}^{N} p_i \frac{x_i}{r} \log_2 \frac{x_i}{r},$$  [1]

where $p_i = \dfrac{t_i}{\sum\limits_{i=1}^{N} t_i}$ was the occupancy probability, $r = \sum\limits_{i=1}^{N} p_i x_i$ was the mean firing rate, and $N$ was the number of bins/grids. Trajectory SI was given by *Equation [1]* with $x_i$ and $t_i$ computed using the neuron's spikes and the animal's occupancy time during all laps of a trajectory. For lap SI, we first obtained SI for each single lap according to *Equation [1]* with the single lap's rate curve and occupancy times. Lap SI was the mean of all laps' SI values on a trajectory. SI on the 2D open box (open SI) was similarly given by *Equation [1]* with the open box session's rate map and occupancy times.

## Rate-stability

Given any two laps of a trajectory, suppose the neuron's firing rates at the $i$-th bin are $x_i$ and $y_i$. The correlation coefficient between these two laps' spatial rate curves was defined as

$$C = \frac{1}{N} \sum_{i=1}^{N} \left(\frac{x_i - \bar{x}}{\sigma_x}\right)\left(\frac{y_i - \bar{y}}{\sigma_y}\right), \qquad [2]$$

where $\bar{x}$ and $\sigma_x$ are the mean and standard deviation of $[x_1, x_2, …, x_N]$ and $\bar{y}$ and $\sigma_y$ are the mean and standard deviation of $[y_1, y_2, …, y_N]$, respectively. We computed $C$ values for all combinations of laps and took their mean as the rate-stability.

## High-order sequence template construction

Because it was impossible to build a template sequence from the neurons' place fields on a trajectory as in previous studies (*Lee and Wilson, 2002*; *Ji and Wilson, 2007*), we relied on pair-wise cross-correlations (CCs). Given a pair of neurons, CC was computed for each lap on a trajectory. We divided the time period of the lap into 100-ms time bins. If the firing rates of the two neurons were $[x_1, x_2, …, x_N]$ and $[y_1, y_2, …, y_N]$, where $N$ was the number of time bins, the correlation coefficient $C(\Delta T)$ at time lag $\Delta T$ was given by,

$$C(\Delta T) = \frac{1}{N} \sum_{i=1}^{N} \left(\frac{x_i - \bar{x}}{\sigma_x}\right)\left(\frac{y_{i+\Delta T} - \bar{y}}{\sigma_y}\right). \qquad [3]$$

The procedure for deriving a template sequence from CCs is the following. We assigned letter identifications to active neurons on a trajectory. For example, if eight neurons were active, we assigned them to *A B C D E F G H*. Then, we determined whether any pairs of neurons had a stable CC and if so, we used their CC peak times to order the neurons within the pairs (see *Figure 3—figure supplement 1* for details). Next, we derived a sequence that agreed with all the ordered pairs. For example, if ordered pairs were found to be *AB, FA, BH, BD, HD*, we arrived at a sequence *FABHD*. However, it was not always as straightforward as this, because of the following complications. (1) Sometimes contradictory orders, for example *AB, BC, CA* occurred, due to the noisy nature of spike trains, especially when multiple neurons fired close in time. (2) There were often many undetermined orders, because not all pairs were found to have a stable CC. This resulted in too much freedom in template sequences and too many different sequences could agree with the ordered pairs to the same degree (as defined below).

Our strategy was to use the following algorithm to derive a template sequence from all ordered pairs of a trajectory:

1. Determine the number of neurons ($k$) that participated in the ordered pairs. If $k \leq 10$, generate all possible sequences of these neurons. Otherwise, randomly generate 500,000 sequences.
2. Search through these initial sequences to identify the best sequences, which are those agree with the ordered pairs the most. The degree of agreement is determined by an ordered-pair score $s$: the percentage of all ordered pairs that match the orders in the sequence.
3. Check if the best sequences meet the template criteria: First, only one best sequence is resulted or multiple equivalent best sequences co-exist only because some ordered pairs have peaks at 0 s. Second, they need to satisfy either (i) $s > 80\%$ and the all-pair score $q > 90\%$, where $q$ is the percentage of all possible pairs in the sequence that match with the ordered pairs, or (ii) the nearest-neighbor score $b > 90\%$, where $b$ is the percentage of all nearest neighbor pairs in the sequence that match with the ordered pairs.
4. If the template criteria are met, a template sequence is obtained (if multiple equivalent sequences exist, randomly pick one) and the process is terminated. If not, eliminate from the best sequences

and the ordered pairs the neuron that had stable relationship with the least number of other neurons. Use the remaining best sequences as initial sequences and repeat from step (2).

5. If the number of remaining neurons is less than four, the process is terminated and no template is obtained.

Given that there were insufficient numbers of active neurons on some trajectories and that many neurons did not have stable CCs, not all trajectories resulted in a successful template sequence. The template sequences derived and their corresponding trajectories and animals are listed in *Tables 2 and 3*.

## Template sequence matching

We modified our previously published method (*Ji and Wilson, 2007*) to search for firing sequences that matched with the template on a track trajectory in the spike patterns of all individual laps of the trajectory and in open box sessions (see details in *Figure 3—figure supplement 2*). Briefly, we transformed a multi-cell spike pattern into a long sequence, according to each neuron's firing peak times. We then identified segments of the sequence that matched with the template by a rank-order correlation.

To illustrate the detected sequences, we arranged neurons according to their orders in the template and plotted their spike patterns in the spatial (vs positions on a trajectory, e.g., as in *Figure 3A*) or temporal (vs time, e.g., as in *Figure 5A*) domain. For illustration purposes only, peak times/positions of the neurons within each detected sequence were linear-regressed with their ranks in the template. The result was plotted as an angled line to highlight the sequence.

The absolute number of detected matches with a template sequence depends on the number of neurons in the template, the parameters that transform spike patterns to sequences, and how strict the rank-order correlation criterion is. Therefore, we expressed the number of detected sequences ($n$) as a $Z$-score relative to the chance-level, determined by the following shuffling procedure (*Ji and Wilson, 2007*). We randomly shuffled the neurons' identities and generated 1000 copies of shuffled spike patterns. For each copy, we counted the number of sequences matched with the template. We computed the mean ($m$) and standard deviation ($\sigma$) of the counts for all the copies. The $Z$-score of $n$ is $Z = (n - m)/\sigma$. The number $n$ was considered significant if $Z > 1.645$ (corresponding to p<0.05, $Z$-test). This $Z$-score normalization was a robust measurement of number of sequences, insensitive to the particular choice of a key parameter in the template generation process (*Figure 3—figure supplement 4*).

## Sequence location shift

For each detected matching sequence, we mapped the animal's locations on the trajectory between the start and end times of the sequence and computed the center between the start and end locations as the sequence's center location. To quantify the location shift between any two laps, for each sequence in one lap we computed the distance between its center location and the center location of the nearest sequence in the other lap. If multiple sequences existed in any of the two laps, the distance was computed for every sequence and then averaged as the location shift between the two laps. The location shifts were computed as such for all combinations of laps and their mean value was determined as the location shift for the template sequence on its trajectory.

## Sequence auto-correlation

For all sequences identified as matches with a template during a track running and open box session, we computed a temporal and a spatial auto-correlation to examine the periodicity of the sequence occurrence. For each sequence, we computed the time and the cumulative travel distance of the animal at the center of the sequence. We treated the times (distances) of all the sequences in a session as a 'spike' train and computed its binned rates [$x_1, x_2, …, x_N$] with a temporal (spatial) bin size of 2 s (4 cm). The auto-correlation was computed using *Equation [3]* with two identical vectors ($x_i = y_i, i = 1, 2, … N$). The auto-correlogram (auto-correlations plotted against time/space lags) was only computed for those templates or open box sessions that yielded significant number of detected sequences ($Z$-score > 1.645, p<0.05, $Z$-test).

## Power spectral analysis of LFPs

For each raw LFP trace during a track running and an open box session, we computed its power spectral density (PSD) at every 0.5 Hz by a multitaper method using Matlab. The absolute theta power was the

integration of PSD values within the theta band (6–12 Hz). This frequency range was chosen because most PSDs in our data showed a peak centered at 8–10 Hz and spanning this range. The PSD values were normalized by the total power at frequencies >2 Hz. The power at ≤2 Hz was not included in the normalization because it was sensitive to the animal's movement artifacts. The normalized theta power was the integration of the normalized PSD values within 6–12 Hz. The analysis was also performed on LFPs restricted within the start and end times of identified firing sequences.

## Acknowledgements

We thank H-C Lu for helping set up the mouse colony, Y Dabaghian, C Kemere, H-C Lu, H Zheng and Ji lab members for helpful discussions, and R Davis, H Zoghbi and H Penagos for critical readings of the manuscript.

## Additional information

### Funding

| Funder | Grant reference number | Author |
| --- | --- | --- |
| National Institute of Neurological Disorders and Stroke | R21NS075726 | Daoyun Ji |

The funder had no role in study design, data collection and interpretation, or the decision to submit the work for publication.

### Author contributions

JC, Acquisition of data, Analysis and interpretation of data; DJ, Conception and design, Analysis and interpretation of data, Drafting or revising the article

### Ethics

Animal experimentation: All procedures and experiments described in this study followed the guidelines by the National Institute of Health and were approved by the Institutional Animal Care and Use Committee at Baylor College of Medicine (Animal protocol number: AN-5134).

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
