## [Decision Letter]

Thank you for sending your work entitled “Rigid firing sequences undermine spatial memory codes in a neurodegenerative mouse model” for consideration at *eLife*. Your article has been favorably evaluated by a Senior editor and 3 reviewers, one of whom is a member of our Board of Reviewing Editors.

The Reviewing editor and the other reviewers discussed their comments before we reached this decision, and the Reviewing editor has assembled the following comments to help you prepare a revised submission.

This is an outstanding and valuable study that provides new evidence on two major issues: A) the findings show that firing sequences of hippocampal principal cells are independent of the external stimuli (locations and other cues) to which they can become bound with experience; B) and the results highlight that, in an AD mouse model, the sequence structure is retained, but the ability to bind the sequences to behavior and location is diminished. The methods are largely straightforward recording and analysis methods for place cells. The analyses for sequence coding are creative and logical, and nicely controlled for a comparison to random sequences. So, the methods are appropriate, and the contrast between poor spatial coding and strong and rigid sequence coding is compelling.

1) The authors state in the Introduction: “on any given trajectory HP either retrieves stored internal sequences or forms new ones…” Are there data that support this statement? Are these two processes necessarily mutually exclusive, especially on the large scale of a “trajectory”? If not, this should be modified.

2) The data suggest that the rigid sequences in the mutants are not spatially anchored and thus repeat themselves independent of position. Do the current data allow the authors to estimate the spatial period of this repetition? If so, is there a fixed distance across environments and mice?

3) The last section of the Results deals with sex differences found in the mutants (males were worse than females). This is surprising given the 2011 Yao et al papers (Neurobio Aging) that found females had greater memory impairments and increased hyperphosphorylation of Tau compared to males. This discrepancy should be discussed.

4) Is there any evidence of what type of neurons are lost in the mutant’s CA1 (inhibitory vs pyramidal cells)? If so, how does this color your interpretation of the data?

5) The firing sequence templates used in the study are time templates, not spatial templates, detected via an algorithm based on cell-pair cross-correlation. This algorithm is actively searching for the temporally rigid sequences and we have two related concerns. First, the algorithm selects for animals and sessions that have a better representation of temporal sequences and detects 13 templates in 6 WT mice (2.1/animal) and 8 templates in 8 Tau mice (1/animal), twice more in WT. Given that detection of templates has a stage that eliminates neurons that are not reliably cross-correlated with other neurons, we wonder whether this method artificially enhances the detection of rigid sequences by excluding neurons and consequently sessions from the analysis. Second, the neurons of Tau mice seem to fire over more extended periods of time, while animals explore spatial environments which could give more opportunities for short sequences of 4–5 neurons to be detected, particularly given the case that multiple peaks from CA1 neurons are being considered in the template construction. The authors should address these concerns and show that their method does not artificially enhance the detection of sequences in Tau mice.

6) The authors should carefully evaluate/discuss which brain areas are affected in the Tau transgenic mice. Alzheimer's disease affects the cholinergic system and the entorhinal cortex, two areas crucial for theta generation and place field expression. Are entorhinal cortex and medial septum affected in these transgenics? The authors should quantify the theta oscillation in WT, Tau, and additional control animals. What does the EEG activity look like during the expression of the rigid sequences in the Tau mice? Related to this, how is the overall activity of CA1 interneurons in the Tau mice? The authors mention that the rigid sequences could be generated in the CA3. Is the CA3 area spared in the Tau animals?

7) Is there any regularity in the occurrence of the rigid sequences in time?

8) Aside from movement velocity, differences in the overall behavior of the Tau mice could potentially affect the results. Tau animals could be more confused and turn around more, which would preserve temporal sequences but would affect place cell activity. The authors should analyze animals’ trajectories on the linear tracks and show there are no differences between genotypes.

---

## [Author Response]

*1) The authors state in the Introduction: “on any given trajectory HP either retrieves stored internal sequences or forms new ones…” Are there data that support this statement? Are these two processes necessarily mutually exclusive, especially on the large scale of a “trajectory”? If not, this should be modified*.

This sentence has been re-phrased to reflect that this is a theoretical hypothesis and the two types of sequences could be mixed.

*2) The data suggest that the rigid sequences in the mutants are not spatially anchored and thus repeat themselves independent of position. Do the current data allow the authors to estimate the spatial period of this repetition? If so, is there a fixed distance across environments and mice*?

We have added an analysis to examine the spatial and temporal periodicity of the detected sequences. The results (Figure 8) do not reveal any periodicity other than the expected periodic occurrence of WT sequences at a spatial period equal to the track length.

*3) The last section of the Results deals with sex differences found in the mutants (males were worse than females). This is surprising given the 2011 Yao et al papers (Neurobio Aging) that found females had greater memory impairments and increased hyperphosphorylation of Tau compared to males. This discrepancy should be discussed*.

We are grateful for the reference provided (Yue et al., Neurobiol Aging, 2011). We have added the reference and the corresponding paragraph in the Discussion section has been modified. This is a very important reference that provides direct evidence for a sex difference in the transgenic rTg4510 mice. We propose two possible reasons for the discrepancy between this reference and our results. (1) The mice in our study were much older than those in the reference. (2) Tau pathology may affect memory-unrelated factors, such as stress, in female rTg4510 mice more than those in males.

*4) Is there any evidence of what type of neurons are lost in the mutant's CA1 (inhibitory vs pyramidal cells)? If so, how does this color your interpretation of the data*?

There is evidence that the CamKII gene used to control the transgene expression is not expressed in interneurons. Therefore, although it has not been explicitly studied, the tau pathology and cell death processes are expected to take place only in pyramidal neurons. However, this does not necessarily mean that there would be no changes in interneurons. There could be still adaptive changes. We have computed the firing rates of the limited number of interneurons in our datasets to show that interneurons in Tau mice fired with a lower rate. This type of adaptive change could maintain the overall firing rate of remaining pyramidal neurons at a relatively normal level. The interneuron firing rate result and related discussion are included in the Discussion section, because this is a side result and it is appropriate for the topic at place.

*5) The firing sequence templates used in the study are time templates, not spatial templates, detected via an algorithm based on cell-pair cross-correlation. This algorithm is actively searching for the temporally rigid sequences and we have two related concerns. First, the algorithm selects for animals and sessions that have a better representation of temporal sequences and detects 13 templates in 6 WT mice (2.1/animal) and 8 templates in 8 Tau mice (1/animal), twice more in WT. Given that detection of templates has a stage that eliminates neurons that are not reliably cross-correlated with other neurons, we wonder whether this method artificially enhances the detection of rigid sequences by excluding neurons and consequently sessions from the analysis. Second, the neurons of Tau mice seem to fire over more extended periods of time, while animals explore spatial environments which could give more opportunities for short sequences of 4–5 neurons to be detected, particularly given the case that multiple peaks from CA1 neurons are being considered in the template construction. The authors should address these concerns and show that their method does not artificially enhance the detection of sequences in Tau mice*.

These are very important concerns. Since we do not know which sequences are “real”, it is hard to measure the “enhancement” of sequence detection. Therefore, we addressed the concerns by analyzing how our sequence detection results depend on (A) the threshold used in the cell-pair selection and (B) the temporal overlap of spikes of multiple cells (especially in Tau mice). The results of this analysis have been presented in Figure 3–figure supplement 4. As the threshold is reduced from a stringent to a low, more relaxed value for cell-pair selection, the number of templates is expectedly increased. However, the mean number of detected sequences (in Z-score) is hardly changed for a wide range of threshold values. Only when the threshold is very low (< 0.08) does the Z-score start to change significantly. Second, to probe the effect of temporal overlap of spiking activities, we binned each spike train in 1-s bins and eliminated spikes within randomly chosen bins. We then computed the number of detected sequence (in Z-score) with the down-sampled spike patterns. The down sampling is to systematically vary the level of spiking overlaps among neurons at a time scale of the sequences detected. We found that the resulted Z-score changes very slowly as more and more spikes are removed. Even with only 40% spikes remaining, the mean Z-scores of WT and Tau mice do not significantly change from the values computed using the original spike patterns. Therefore, although the number of templates and the number of detected sequences inevitably vary with the particular threshold chosen and the specific details in the firing patterns, the normalization (Z-scoring relative to the shuffle-generated number of sequences) provides a robust measure of the sequential structure in the multicell spiking patterns.

*6) The authors should carefully evaluate/discuss which brain areas are affected in the Tau transgenic mice. Alzheimer's disease affects the cholinergic system and the entorhinal cortex, two areas crucial for theta generation and place field expression. Are entorhinal cortex and medial septum affected in these transgenics? The authors should quantify the theta oscillation in WT, Tau, and additional control animals. What does the EEG activity look like during the expression of the rigid sequences in the Tau mice? Related to this, how is the overall activity of CA1 interneurons in the Tau mice? The authors mention that the rigid sequences could be generated in the CA3. Is the CA3 area spared in the Tau animals*?

(A) We have now added to the Discussion that entorhinal cortex is affected by the pathology more than CA3. The medial septum has not been specifically examined, but should be affected based on the expression pattern of CamKII-tTA (Odeh, et al., Neuroimage, 2011). [Our own histology data could not confirm this without the staining of identified septal cholinergic neurons using antibodies specific to phosphorylated tau.] (B) In the Discussion, we have added that interneurons in Tau mice have lower firing rates than those in WT mice. See the response to point #4. (C) We have now quantified theta oscillations in WT and Tau mice (Figure 10). We found that prominent theta is still present in the local field potentials (LFPs) of Tau mice. Although the absolute LFP power is lower in Tau mice, as expected by the cell loss, the proportion of theta power is comparable between WT and Tau mice. Restricting the analysis to LFPs within rigid sequences returned a similar result. (D) Based on the result of EEG analysis and the structural differences in the pathology, we propose that CA3 plays a major role in driving the rigid sequences in Tau mice.

*7) Is there any regularity in the occurrence of the rigid sequences in time*?

We have examined the temporal periodicity of Tau sequences and found no sign of periodic occurrence of Tau sequence in time (Figure 8). See also the response to point #2.

*8) Aside from movement velocity, differences in the overall behavior of the Tau mice could potentially affect the results. Tau animals could be more confused and turn around more, which would preserve temporal sequences but would affect place cell activity. The authors should analyze animals’ trajectories on the linear tracks and show there are no differences between genotypes*.

Besides number of laps completed per session and running speed, we have added several additional parameters to quantify the animals’ trajectories: the total travel distance per lap (which captures the turning around behavior), the number of stops per lap, and the mean duration of stopping per lap. The results, as well as example raw running trajectories, are plotted in Figure 9. The total travel distance and stopping duration were found to be similar between WT and Tau mice.